# Structured Multi-modal Graph Disentanglement for Psychiatric Diagnosis

**Hongyu Shi** [* 1] **Kaizhong Zheng** [* 1] **Wensheng Zhai** [1] **Shuai Jiang** [1] **Badong Chen** [1] **Liangjun Chen** [1]

## Abstract

Multi-modal neuroimaging-based psychiatric diagnosis must integrate cross-modal agreement with modality-specific complementarity, yet in real multi-site cohorts these signals are frequently entangled with site- and cohort-dependent correlations, yielding shortcut-driven predictions and limited interpretability. We propose Structured Multi-modal Graph Disentanglement (SMGD), which explicitly factorizes multi-modal graph representations into four components with distinct roles: shared diagnostic evidence, complementary diagnostic evidence, incidental cross-modal agreement, and modality-specific non-robust correlations, with the former two forming the diagnostic core and the latter two suppressed as shortcuts. SMGD is realized as geometry-driven structure learning: under a mild distributional assumption, we develop mini-batch estimable surrogate regularizers that shape subspace organization and cross-modal relations, enforcing semantic consistency through relational geometry rather than centroid coincidence while suppressing confounded dependencies. Experiments on large multi-site datasets show improved in-domain diagnosis and more reliable cross-dataset generalization in the presence of a modality gap, without relying on expert-crafted diagnostic biomarkers.

## 1. Introduction

The diagnosis of complex psychiatric disorders, such as Autism Spectrum Disorder (ASD) and Schizophrenia Spectrum Disorder (SSD) has increasingly relied on the integration of multi-modal neuroimaging data in recent years. This paradigm shift is driven by the fact that biological substrates

---

[*]Equal contribution [1]State Key Laboratory of Human-Machine Hybrid Augmented Intelligence, Institute of Artificial Intelligence and Robotics, Xi'an Jiaotong University, China. Correspondence to: Liangjun Chen <liangjunchen@xjtu.edu.cn>, Kaizhong Zheng <kzzheng@xjtu.edu.cn>.

*Proceedings of the 43rd International Conference on Machine Learning*, Seoul, South Korea. PMLR 306, 2026. Copyright 2026 by the author(s).

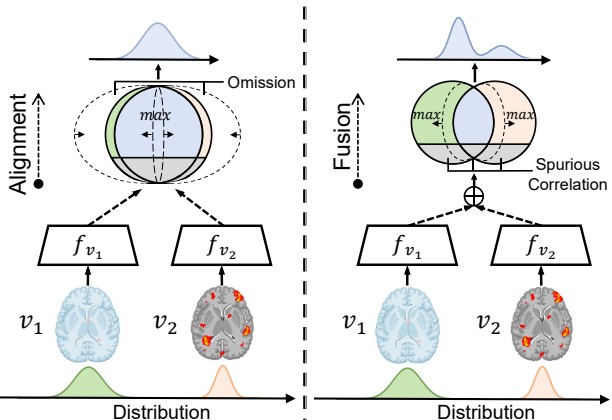

*Figure 1.* Illustration of alignment and fusion paradigms. Alignment maximizes cross-modal mutual information but may omit modality-specific cues, whereas fusion aggregates multi-modal information for prediction, but may also capture spurious correlations. ($v_1, v_2$: modalities; $f_{v_i}$: modality-specific encoders; gray regions indicate non-diagnostic or spurious factors).

are intrinsically coupled. For instance, determining whether atypical functional connectivity is associated with underlying structural degeneration or pure functional dysregulation requires the simultaneous analysis of both modalities (Calhoun & Sui, 2016). The integration of these heterogeneous modalities promises to reveal robust biomarkers missed by single-modal analyses, highlighting the importance of multi-modal learning in modern medical image analysis.

As illustrated in Figure 1, existing multi-modal diagnosis pipelines primarily leverage two key strategies: alignment and fusion (Sui et al., 2023; Li & Tang, 2024). Alignment methods often map modalities into a shared embedding space and use contrastive objectives to increase cross-modal agreement or mutual-information-based dependence between paired samples (Oord et al., 2018; Radford et al., 2021; Ma et al., 2024). While effective for learning modality-invariant semantics, aggressive alignment may omit important modality-specific cues. More critically, the persistent geometric separation denoted as the modality gap shows that even with contrastive alignment, modality embeddings can remain separated, indicating residual modality-specific nuisances that cannot be fully resolved by pairwise matching (Liang et al., 2022; Zhang et al., 2024b; An et al., 2025; Almudévar et al., 2025). Fusion methods, on the other hand, aggregate multi-modal features via concatena-

tion, attention, or graph-based aggregation to preserve both consensus and complementarity (Zadeh et al., 2017; Cui et al., 2022; Kan et al., 2022; Xu et al., 2023). However, fusion faces challenges from latent entanglement: while consensus and complementarity can provide predictive cues, they may also absorb non-diagnostic factors (e.g., demographics and noise), leading to confounded representations and reduced reproducibility across datasets (Huang et al., 2025; Wu et al., 2025). Without explicit partitioning or constraints, models may overfit cohort-specific shortcuts, making fusion gains brittle under distribution shifts (Yang et al., 2024).

Recent studies have further developed these paradigms by introducing explicit disentanglement and regularization mechanisms that refine how alignment and fusion are carried out and clarify what geometry the latent space should satisfy. Specifically, several works incorporate information bottleneck, dual-objective learning, and common information theory to isolate and regulate heterogeneous components within multi-modal representations (Zhang et al., 2024a; Almudévar et al., 2025; Huang et al., 2025). A unifying view is to maximize predictive information between the learned representation $\mathbf{Z}$ and the diagnostic target $Y$ ($I(\mathbf{Z}; Y)$), while enforcing a structural constraint $\mathcal{R}_{struct}(\mathbf{Z})$, where different choices of $\mathcal{R}_{struct}$ induce different partitions of multi-modal information. Despite this progress, existing instantiations of $\mathcal{R}_{struct}$ remain relatively fragmented, often emphasizing only a subset of structural properties without jointly coordinating them, which can leave the representation only partially organized when modality heterogeneity and cohort-specific confounds are entangled.

To resolve these limitations, we propose Structured Multi-modal Graph Disentanglement (SMGD), a fine-grained factorization framework tailored to psychiatric diagnosis that separates actionable diagnostic evidence from non-robust correlations in multi-modal graph representations. For each modality, SMGD decomposes the embedding into four subspaces: shared diagnostic ($Z_S$), complementary diagnostic ($Z_C$), incidental consensus ($Z_I$), and modality-specific nuisances ($Z_N$). Prediction is driven by the diagnostic core ($Z_S, Z_C$), while ($Z_I, Z_N$) are treated as shortcut channels and are discouraged from carrying label-discriminative signals. Crucially, this decomposition is learned through joint diagnosis-and-modality objectives: SMGD preserves cross-modal structure within the shared subspaces ($Z_S, Z_I$) while promoting cross-modal independence for modality-private subspaces ($Z_C, Z_N$). Under a mild statistical assumption, these goals are enforced with simple batch-wise geometric regularizers, yielding a structured representation that is less prone to shortcut learning and more stable under distribution shifts. The key contributions are summarized as follows:

- Cast multi-modal graph psychiatric diagnosis as struc-

tured information learning and reconcile the objectives of alignment and fusion through a quadruple latent factorization that separates diagnostic evidence from incidental agreement and modality-specific nuisances.

- Develop and train the SMGD framework with modality-specific encoders and a disentanglement module, and, under a mild distributional assumption, introduce geometric surrogate objectives that jointly constrain the structure of the designated latent subspaces.

- Validate SMGD on multi-site datasets, showing improved within-dataset diagnosis and stronger cross-dataset generalization, with ablations indicating consistent gains from the proposed structural decomposition and regularization.

## 2. Related Work

### 2.1. Multi-modal Representation Learning

Multi-modal diagnosis typically combines representation learning that encourages cross-modal compatibility with downstream fusion for prediction (Sui et al., 2023; Li & Tang, 2024). A prominent line of work adopts contrastive learning (Radford et al., 2021; Ma et al., 2024) to increase agreement between modalities, often formulated as maximizing cross-modal mutual information to induce a shared semantic space. While effective at extracting modality-invariant cues, such objectives may over-emphasize commonality and under-represent modality-specific evidence, especially in the presence of a non-negligible modality gap (Liang et al., 2022; Almudévar et al., 2025). Building on these representations, fusion models integrate multi-modal features through tensor fusion (Zadeh et al., 2017), attention-based aggregation (Xu et al., 2023), or brain-tailored graph architectures (Cui et al., 2022; Kan et al., 2022) to exploit both consensus and complementarity. However, without explicit mechanisms to control how nuisance factors propagate through the fused space, diagnostic signals can be mixed with noise and incidental correlations.

### 2.2. Disentanglement for Brain Imaging

Disentanglement in brain imaging aims to separate latent representations into clinically or anatomically meaningful factors, such as disease-relevant patterns, modality-specific variations, and nuisance factors arising from acquisition or cohort heterogeneity. Various methods have been proposed to achieve this goal: anatomical and modality information is explicitly disentangled in multi-modal brain magnetic resonance imaging (MRI) by introducing margin-based regularization and modality-aware encoding (Ouyang et al., 2021). CI-GNN learns disentangled representations by applying a conditional mutual information constraint, enabling the separation of causal and non-causal subgraph-level pat-

terns in brain networks (Zheng et al., 2024). Along this line, MGDR disentangles multi-modal graph representations into common and modality-private components for brain disease prediction, aiming to jointly leverage shared and complementary information across modalities (Jiang et al., 2024). More recently, geometric constraints inspired by self-supervised learning have been increasingly adopted as disentanglement drivers to promote statistical decoupling. Barlow Twins reduces redundancy by pushing the cross-correlation matrix toward the identity (Zbontar et al., 2021). VICReg combines invariance with explicit variance and covariance regularization to prevent representational collapse while suppressing feature redundancy (Bardes et al., 2022). Crocodile proposes a cross-experts covariance loss that directly penalizes covariance between expert-specific embeddings to encourage disentangled factors (Lin et al., 2025). DAE enforces decorrelation in the latent space to decouple factors under deterministic autoencoders (Cha & Thiyagalingam, 2023).

## 3. Motivation

### 3.1. Problem Formulation

Let $\mathcal{D} = \{(\mathcal{G}_m, Y_m)\}_{m=1}^M$ denote a dataset comprising $M$ subjects, where $Y_m \in \mathcal{Y}$ represents the diagnostic label. For each subject, we observe a set of multi-modal graphs $\mathcal{G}_m = [\mathcal{G}_m^{(v)}]_{v=1}^V$. Following standard graph construction, each view-specific graph is represented as $\mathcal{G}_m^{(v)} = [A_m^{(v)}, X_m^{(v)}]$, where $A_m^{(v)} \in \mathbb{R}^{N \times N}$ is the adjacency matrix over $N$ graph nodes, and $X_m^{(v)} \in \mathbb{R}^{N \times d_x}$ is the node feature matrix with $d_x$-dimensional features for each node.

We employ modality-specific encoders $E_{\theta_v}$ to map each view to a latent representation $Z^{(v)} = E_{\theta_v}(\mathcal{G}^{(v)})$ and denote the collection of multi-view representations as $\mathbf{Z} = [Z^{(v)}]_{v=1}^V$. Our goal is to learn a representation space that is both discriminative and structurally regularized. Formally, we define this goal as a constrained optimization problem:

$$\max_{\{\theta_v\}} \underbrace{I(\mathbf{Z}; Y)}_{\text{Prediction}}, \quad \text{s.t.} \quad \mathcal{R}_{struct}(\mathbf{Z}) \leq \varepsilon, \qquad (1)$$

where $I(\mathbf{Z}; Y)$ quantifies the predictive information relevant to the task and $\varepsilon$ is a tolerance threshold, the term $\mathcal{R}_{struct}(\cdot)$ serves as a general structural constraint functional that regularizes the geometric or statistical properties of the latent space. Many learning paradigms can be viewed as specific instantiations of this constraint: for example, Information Bottleneck (IB) minimizes input redundancy by constraining $I(\mathbf{Z}; \mathcal{G})$, while contrastive learning encourages cross-modal alignment by maximizing lower bounds on the mutual information between paired modality-specific representations, or equivalently by constraining a negative mutual-information surrogate.

### 3.2. Structured Multi-modal Graph Disentanglement

To provide a rigorous account of multi-modal interactions in psychiatric diagnosis, we need to explicitly characterize heterogeneous information components in the latent space. While Partial Information Decomposition (PID) partitions the label-relevant mutual information into redundancy ($I_\cap$), uniqueness ($U_v$), and synergy ($I_{\text{syn}}$) (Bertschinger et al., 2014; Liang et al., 2023a;b; Dewan et al., 2024; Zhao et al., 2025), it does not distinguish actionable diagnostic evidence from shortcut-like correlations, nor does it explicitly characterize how such non-diagnostic dependencies interact across modalities.

We therefore propose Structured Multi-modal Graph Disentanglement (SMGD), a framework that refines latent representations by explicitly isolating decision-worthy diagnostic evidence from shortcut-like correlations. Instead of a coarse separation, SMGD performs a fine-grained factorization of each view-specific representation into four semantically distinct subspaces:

$$Z^{(v)} = [Z_S^{(v)}, Z_C^{(v)}, Z_I^{(v)}, Z_N^{(v)}]. \qquad (2)$$

This decomposition captures the heterogeneity of multi-modal information based on two intrinsic properties: cross-modal consistency and diagnostic relevance. Specifically, $Z_S$ (Shared) and $Z_C$ (Complementary) constitute the diagnostic core, aggregating actionable evidence for prediction. In contrast, $Z_I$ (Incidental) and $Z_N$ (Nuisance) collect non-actionable correlations that are statistically discouraged from contributing to the decision path. For clarity, we instantiate the formulation with two modalities, denoted by $v_1$ and $v_2$. The corresponding definitions are given as follows.

**Definition 3.1** (Diagnostic Sufficiency). Let $Z_{\text{core}} = [Z_S^{(v_1)}, Z_S^{(v_2)}, Z_C^{(v_1)}, Z_C^{(v_2)}]$ denote the diagnostic core. It is sufficient if it satisfies

$$[Z^{(v_1)}, Z^{(v_2)}] \perp\!\!\!\perp Y \mid Z_{\text{core}}, \qquad (3)$$

i.e., $Z_{\text{core}}$ captures all the predictive content needed for diagnosis.

**Definition 3.2** (Shortcut Neutrality). Let $Z_{\text{short}} = [Z_I^{(v_1)}, Z_I^{(v_2)}, Z_N^{(v_1)}, Z_N^{(v_2)}]$ denote shortcut-like components. We aim for approximate diagnostic neutrality by requiring their label-predictive dependence to be small. Specifically, rather than claiming strict independence, which is often unattainable in finite and biased cohorts, we enforce that $Z_{\text{short}}$ remains weakly dependent on $Y$, thereby diminishing its utility as shortcut components:

$$I(Z_{\text{short}}; Y) \leq \varepsilon_s, \qquad (4)$$

where $\varepsilon_s \geq 0$ controls the maximum allowable information leakage. Meanwhile, solely constraining diagnostic

relevance is insufficient to guarantee effective disentanglement. To preserve modality-specific complementary information while enabling cross-modal integration, we introduce targeted structural constraints. Specifically, for shared subspaces $\mathcal{A}_1 \in \{S, I\}$, we avoid standard alignment objectives (e.g., MSE, InfoNCE) because enforcing centroid coincidence can suppress the information of the complementary subspace $Z_C$ (see Appendix B for proof). Instead, we require structural consistency via a relational divergence $\mathcal{D}(\cdot, \cdot)$ that is robust to the modality gap:

$$\mathcal{D}(Z_{\mathcal{A}_1}^{(v_1)}, Z_{\mathcal{A}_1}^{(v_2)}) \leq \epsilon_{m_1}, \tag{5}$$

where $\mathcal{D}(\cdot, \cdot)$ is a relational divergence invariant to isometry and global scaling; $\epsilon_{m_1}$ controls the tolerance of structural consistency.

Conversely, for modality-private subspaces $\mathcal{A}_2 \in \{C, N\}$, we explicitly encourage cross-modal independence:

$$I(Z_{\mathcal{A}_2}^{(v_1)}; Z_{\mathcal{A}_2}^{(v_2)}) \leq \epsilon_{m_2}, \tag{6}$$

where $\varepsilon_{m_2}$ controls the maximum allowable dependence between modality-private components.

## 4. Methodology

The SMGD objectives are defined in terms of mutual information, which lacks closed-form estimators for high-dimensional representations. We establish a principled connection to tractable geometric surrogates via the Data Distribution Hypothesis (DDH) (Shwartz-Ziv et al., 2023).

**Proposition 4.1** (Data Distribution Hypothesis). *Consider a deterministic neural network $f$ with piecewise affine activations (e.g., ReLU). If the input distribution around each data point $x_n^*$ ($n$ indexing the dataset) is a small-variance Gaussian $X|T{=}n \sim \mathcal{N}(x_n^*, \Sigma_{x_n^*})$ whose effective support lies entirely within a single linear region $\omega$ of the network, then $f$ acts as an affine map $f(x) = A_\omega x + b_\omega$ on this support, and the output preserves Gaussianity:*

$$Z|T{=}n \sim \mathcal{N}(A_\omega x_n^* + b_\omega,\ A_\omega \Sigma_{x_n^*} A_\omega^\top). \tag{7}$$

This enables information-theoretic analysis of deterministic networks by shifting randomness from the network to the input, providing a principled basis for subsequent analysis. Specifically, under the Gaussian approximation induced by DDH, the relevant representation variables can be treated as approximately jointly Gaussian within mini-batch estimation. We can then invoke the following well-known equivalence:

**Proposition 4.2** (Gaussian Independence-Covariance Equivalence). *Let $X, Y$ be jointly Gaussian. The mutual information vanishes if and only if the cross-covariance is zero:*

$$I(X;Y) = 0 \iff \mathrm{Cov}(X, Y) = 0, \tag{8}$$

*where $\mathrm{Cov}(\cdot, \cdot)$ denotes the cross-covariance matrix.*

This equivalence motivates approximating the intractable SMGD constraints by shaping the block structure of the joint covariance matrix. We now derive geometric surrogates for each constraint.

### 4.1. Non-degeneracy Constraint

To ensure all four subspaces learn expressive representations, we first aim to maximize their differential entropy $H(Z_{\mathcal{A}})$ for $\mathcal{A} \in \{S, C, I, N\}$ (Shwartz-Ziv et al., 2023). Under DDH, $Z_{\mathcal{A}}$ is approximately Gaussian, and its entropy is proportional to $\log \det(\mathrm{Cov}(Z_{\mathcal{A}}))$. To avoid the numerical instability of computing the determinant directly, we leverage Hadamard's inequality, which states that $\det(\mathrm{Cov}(Z_{\mathcal{A}})) \leq \prod_j \mathrm{Cov}(Z_{\mathcal{A}})_{jj}$, where $\mathrm{Cov}(Z_{\mathcal{A}})_{jj}$ denotes the diagonal entries, with equality holding if and only if $\mathrm{Cov}(Z_{\mathcal{A}})$ is diagonal. Therefore, maximizing entropy can be effectively achieved by jointly maximizing the diagonal variances and minimizing the off-diagonal covariances:

$$\mathcal{L}_{\text{nondeg}}(Z_{\mathcal{A}}) = \sum_{i \neq j} \mathrm{Cov}(Z_{\mathcal{A}})_{ij}^2 - \epsilon_1 \sum_j \log(\mathrm{Cov}(Z_{\mathcal{A}})_{jj}), \tag{9}$$

where $\epsilon_1$ balances the magnitude of the two terms.

### 4.2. Cross-modal Relation Regularization

For shared subspaces $\mathcal{A}_1 \in \{S, I\}$, Eq. (5) requires cross-modal structural consistency. We adopt a normalized Gromov-Wasserstein (GW) formulation that compares metric spaces by matching their relational structures rather than absolute positions (Mémoli, 2011; Gong et al., 2022; Rioux et al., 2024):

$$\mathrm{GW}_2^2(D^{(v_1)}, D^{(v_2)}) = \min_{P \in \Pi(B)} \sum_{i,j,k,l} |D_{ik}^{(v_1)} - D_{jl}^{(v_2)}|^2 P_{ij} P_{kl}, \tag{10}$$

where $D^{(v)} \in \mathbb{R}^{B \times B}$ denotes the pairwise distance matrix among the $B$ mini-batch samples within modality $v$, $\Pi(B)$ denotes the transport polytope with uniform marginals. Since paired multi-modal observations establish a natural one-to-one correspondence, we adopt the identity coupling $P^* = \frac{1}{B} I_B$, under which the GW objective simplifies to $\mathrm{GW}_2^2|_{P=P^*} = \frac{1}{B^2} \|D^{(v_1)} - D^{(v_2)}\|_F^2$, we have:

$$D_{\mathcal{A}_1, ij}^{(v)} = \|z_{\mathcal{A}_1, i}^{(v)} - z_{\mathcal{A}_1, j}^{(v)}\|_2, \tag{11}$$

where $z_{\mathcal{A}_1, i}^{(v)}$ denotes the $i$-th sample's representation in subspace $\mathcal{A}_1$ from modality $v$. To ensure scale-invariance across modalities that may exhibit different dynamic ranges, we apply Frobenius normalization: $\tilde{D}_{\mathcal{A}_1}^{(v)} = \frac{D_{\mathcal{A}_1}^{(v)}}{\|D_{\mathcal{A}_1}^{(v)}\|_F + \epsilon_d}$, where $\epsilon_d$ is a small constant for numerical stability. The

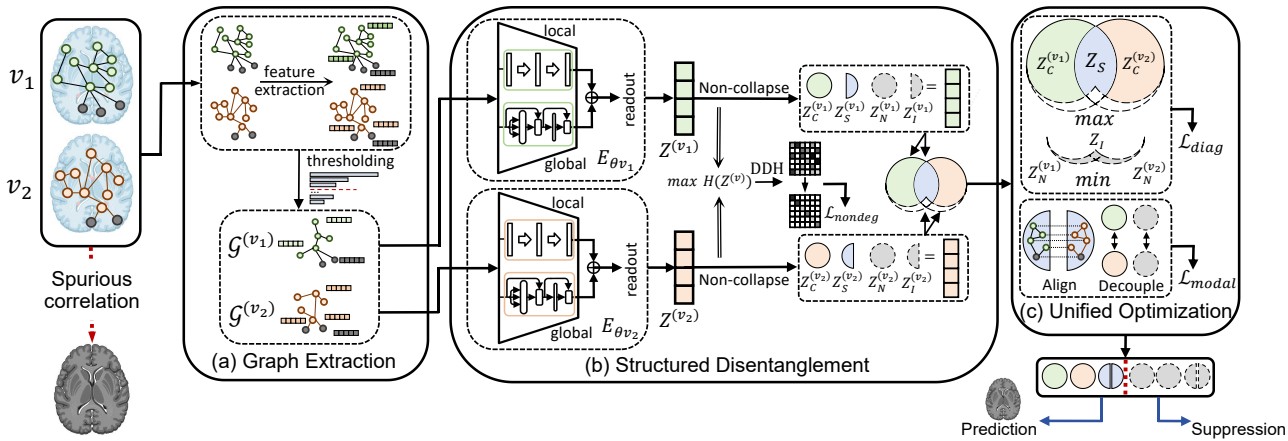

*Figure 2.* **Overview of the SMGD framework.** (a) *Graph Extraction:* for each modality $v \in \{v_1, v_2\}$, node features are extracted and edges are sparsified via thresholding to construct graphs $\mathcal{G}^{(v_1)}$ and $\mathcal{G}^{(v_2)}$ (potentially containing spurious correlations). (b) *Structured Disentanglement:* modality-specific encoders $E_{\theta_1}$ and $E_{\theta_2}$ (with local and global branches) produce embeddings $Z^{(v_1)}$ and $Z^{(v_2)}$, which are further factorized into four subspaces $Z_S^{(v)}, Z_C^{(v)}, Z_I^{(v)}, Z_N^{(v)}$; a non-collapse regularizer maximizes $H(Z_{\mathcal{A}}^{(v)})$ for $\mathcal{A} \in \{S, C, I, N\}$ to prevent degenerate solutions, and DDH motivates covariance-based geometric surrogates. (c) *Unified Optimization:* the diagnostic head predicts $Y$ from the diagnostic core (shared and complementary factors, $Z_S$ and $Z_C$) with loss $\mathcal{L}_{\text{diag}}$, while the cross-modal regularization term $\mathcal{L}_{\text{modal}}$ enforces cross-modal alignment for shared factors and independence for modality-specific factors, helping separate shortcut-related variations from diagnosis.

alignment loss then compares normalized relational structures for shared subspaces:

$$\mathcal{L}_{\text{align}}(Z_{\mathcal{A}_1}) = \frac{1}{B^2} \sum_{i,j} \left( \tilde{D}_{\mathcal{A}_1, ij}^{(v_1)} - \tilde{D}_{\mathcal{A}_1, ij}^{(v_2)} \right)^2. \quad (12)$$

For specific subspaces $\mathcal{A}_2 \in \{C, N\}$, Eq. (6) requires cross-modal independence. Under DDH, this information-theoretic constraint admits a geometric surrogate: encouraging $I(Z_{\mathcal{A}_2}^{(v_1)}; Z_{\mathcal{A}_2}^{(v_2)})$ to be small can be approximated by nullifying the cross-modal covariance (Proposition 4.2):

$$\mathcal{L}_{\text{decoup}}(Z_{\mathcal{A}_2}) = \|\text{Cov}(Z_{\mathcal{A}_2}^{(v_1)}, Z_{\mathcal{A}_2}^{(v_2)})\|_F^2. \quad (13)$$

### 4.3. Diagnostic Information Factorization

To regulate the predictive sufficiency and shortcut neutrality of the learned subspaces, we employ a Lagrangian relaxation to form an unconstrained objective:

$$\max_{\mathbf{Z}} \ I(Z_{\text{core}}; Y) - I(Z_{\text{short}}; Y), \quad (14)$$

where $Z_{\text{core}} = [Z_S^{(v_1)}, Z_S^{(v_2)}, Z_C^{(v_1)}, Z_C^{(v_2)}]$, $Z_{\text{short}} = [Z_I^{(v_1)}, Z_I^{(v_2)}, Z_N^{(v_1)}, Z_N^{(v_2)}]$. Under DDH, both mutual information terms reduce to tractable geometric objectives that admit efficient mini-batch estimation.

For the diagnostic core $Z_{\text{core}}$, maximizing $I(Z_{\text{core}}; Y)$ is achieved through the standard variational lower bound. Since $I(Z_{\text{core}}; Y) \geq H(Y) + \mathbb{E}[\log p_\theta(Y \mid Z_{\text{core}})]$, tightening this bound reduces to minimizing the cross-entropy

loss $\mathbb{E}[-\log p_\theta(Y \mid Z_{\text{core}})]$, which encourages the diagnostic subspace to capture maximally predictive features.

For the shortcut components $Z_{\text{short}}$, the Gaussian structure induced by DDH enables a principled reduction: the constraint $I(Z_{\text{short}}; Y) \leq \varepsilon_s$ can be approximated by a second-order dependence surrogate based on the cross-covariance between representations and labels. This yields a direct geometric surrogate $\|\text{Cov}(Z_{\text{short}}, Y)\|_F^2$, which we estimate via mini-batch statistics:

$$\hat{\text{Cov}}(Z_{\text{short}}, Y) = \frac{1}{B-1} \bar{Z}_{\text{short}}^\top \bar{Y}, \quad (15)$$

where $\bar{Z}_{\text{short}}$ and $\bar{Y}$ denote centered representations and labels, respectively. Minimizing $\|\hat{\text{Cov}}(Z_{\text{short}}, Y)\|_F^2$ drives class-conditional centroids toward coincidence, thereby effectively reducing class-discriminative structure in the shortcut subspace.

### 4.4. Unified Optimization Objective

We unify the diagnosis constraints (Eq. 14) as $\mathcal{L}_{\text{diag}}$, and the modality constraints (Eqs. 12, 13) as $\mathcal{L}_{\text{modal}}$. Together with the non-degeneracy constraint (Eq. 9), the complete SMGD objective is:

$$\mathcal{L}_{\text{total}} = \mathcal{L}_{\text{nondeg}} + \lambda_M \mathcal{L}_{\text{modal}} + \lambda_D \mathcal{L}_{\text{diag}}, \quad (16)$$

where two primary hyperparameters $\lambda_M$ and $\lambda_D$ control the trade-off between structural regularization and diagnostic performance. Internal weighting coefficients are introduced

*Table 1.* Performance comparison on the ABIDE-I and SRPBS-SSD datasets. All metrics are reported as *mean ± standard deviation* across all test folds. The best results are highlighted in **bold**. ASD: Autism Spectrum Disorder, SSD: Schizophrenia Spectrum Disorder.

| Model | ABIDE-I (Controls vs ASD) | | | SRPBS-SSD (Controls vs SSD) | | |
| --- | --- | --- | --- | --- | --- | --- |
| | *ACC* | *F1* | *MCC* | *ACC* | *F1* | *MCC* |
| GCN (ICLR'17) | 0.688±0.034 | 0.647±0.055 | 0.383±0.068 | 0.901±0.019 | 0.922±0.021 | 0.840±0.041 |
| GraphSAGE (NeurIPS'17) | 0.672±0.039 | 0.642±0.037 | 0.352±0.074 | 0.912±0.007 | 0.912±0.006 | 0.825±0.013 |
| GAT (ICLR'18) | 0.706±0.040 | 0.667±0.068 | 0.422±0.076 | 0.925±0.035 | 0.925±0.036 | 0.852±0.070 |
| GIN (ICLR'19) | 0.673±0.051 | 0.662±0.065 | 0.360±0.100 | 0.932±0.026 | 0.931±0.026 | 0.865±0.052 |
| BrainGNN (MedIA'21) | 0.664±0.029 | 0.627±0.080 | 0.334±0.053 | 0.918±0.020 | 0.919±0.019 | 0.838±0.040 |
| IBGNN (MICCAI'22) | 0.715±0.060 | 0.701±0.079 | 0.434±0.120 | 0.943±0.029 | 0.948±0.028 | 0.899±0.057 |
| FBNetGen (MIDL'22) | 0.695±0.041 | 0.663±0.058 | 0.396±0.083 | 0.922±0.033 | 0.919±0.042 | 0.852±0.057 |
| CycGAT (MICCAI'24) | 0.664±0.034 | 0.627±0.058 | 0.337±0.068 | 0.912±0.036 | 0.907±0.042 | 0.828±0.070 |
| BrainOOD (ICLR'25) | 0.682±0.027 | 0.645±0.056 | 0.375±0.056 | 0.908±0.025 | 0.913±0.021 | 0.823±0.041 |
| STAGIN (NeurIPS'21) | 0.655±0.049 | 0.638±0.065 | 0.317±0.101 | 0.922±0.033 | 0.920±0.040 | 0.849±0.062 |
| BioBGT (ICLR'25) | 0.700±0.050 | 0.705±0.039 | 0.410±0.096 | 0.908±0.044 | 0.914±0.040 | 0.824±0.082 |
| DisenGCN (ICML'19) | 0.669±0.045 | 0.641±0.067 | 0.341±0.096 | 0.942±0.017 | 0.944±0.017 | 0.889±0.035 |
| GIB (NeurIPS'20) | 0.713±0.019 | 0.682±0.053 | 0.439±0.038 | 0.932±0.018 | 0.931±0.022 | 0.866±0.034 |
| DGCL (NeurIPS'21) | 0.710±0.033 | 0.701±0.042 | 0.424±0.066 | 0.939±0.035 | 0.942±0.032 | 0.883±0.068 |
| CI-GNN (NN'24) | 0.697±0.029 | 0.682±0.042 | 0.394±0.061 | 0.932±0.043 | 0.933±0.043 | 0.866±0.085 |
| **SMGD (Ours)** | **0.748±0.034** | **0.725±0.053** | **0.507±0.068** | **0.954±0.032** | **0.957±0.024** | **0.906±0.051** |

to normalize the magnitude of constituent terms within each loss component. Detailed hyperparameter configurations and sensitivity analysis are provided in Appendix F.

# 5. Experiments

## 5.1. Experimental Settings

**Datasets and Graph Construction.** We evaluate on two multi-site resting-state fMRI datasets: ABIDE-I (516 ASD, 555 Controls) and SRPBS, from which we select SRPBS-ASD (125 ASD, 125 Controls) and SRPBS-SSD (147 SSD, 147 Controls) sub-cohorts (Di Martino et al., 2014; Tanaka et al., 2021). Using the Brainnetome Atlas (Fan et al., 2016), we parcellate each brain into 246 regions of interest (ROIs) and construct dual-modal graphs: functional connectivity (FC) graphs from Fisher $z$-transformed Pearson correlations of fMRI time series, and structural similarity (SS) graphs derived from Wasserstein-distance-based similarities of gray matter distributions (Leming et al., 2021). Formally, each subject is represented as a graph $G_m = [A_m^{(v)}, X_m^{(v)}]_{v=1}^V$. The node feature matrix $X_m^{(v)} \in \mathbb{R}^{N \times N}$ is the unthresholded connectivity/similarity matrix, representing the whole-brain connectivity profile of each ROI. The sparse adjacency matrix $A$ retains the top 20% strongest edges (largest absolute correlations or structural similarities) with edge weights normalized to $[0, 1]$. More details are provided in Appendix D.

**Pipeline Architecture.** Overall, SMGD adopts a dual-stream modality-specific graph encoding architecture, followed by a hierarchical disentanglement head and a di-

agnostic classifier with geometry-guided subspace regularization. As illustrated in Figure 2, for each modality $v$, a modality-specific *General, Powerful, Scalable (GPS) Graph Transformer* (Rampášek et al., 2022) encoder $E_{\theta_v}$, which integrates local message passing and global self-attention, maps $G^{(v)}$ to a latent representation $Z^{(v)}$. A hierarchical disentanglement head factorizes $Z^{(v)}$ into $[Z_S^{(v)}, Z_C^{(v)}, Z_I^{(v)}, Z_N^{(v)}]$ via coarse-to-fine splits with projection and pooling. For prediction, we concatenate $[Z_S^{(v_1)}, Z_S^{(v_2)}, Z_C^{(v_1)}, Z_C^{(v_2)}]$ and feed it to a lightweight classifier, while $[Z_I^{(v_1)}, Z_I^{(v_2)}, Z_N^{(v_1)}, Z_N^{(v_2)}]$ are used only by the geometric regularizers. Further architectural details are provided in Appendix C.

**Implementation Details.** We implement our framework using PyTorch 2.3.0 and PyTorch Geometric 2.5.0. All experiments are conducted on a single NVIDIA RTX 4090 GPU with 24GB memory. A ReduceLROnPlateau learning rate scheduler is used with a reduction factor of 0.6 and patience of 10 epochs (minimum learning rate: $1 \times 10^{-7}$). For reproducibility, we set random seeds for PyTorch, NumPy, and CUDA, and enable deterministic cuDNN operations. For model instantiation, the GPS encoder is built with two layers, a hidden dimension of 128, and 4 attention heads, and the latent dimension of each disentangled component is set to 64. The model is optimized using AdamW with an initial learning rate of $1 \times 10^{-3}$, weight decay of $1 \times 10^{-3}$, and a dropout rate of 0.6. The batch size is set to 32 and the model is trained for 50 or 100 epochs depending on the dataset size. We perform model selection using stratified $k$-fold cross-validation: within each split, the non-test folds

*Table 2.* Cross-dataset generalization performance.

| Model | ACC | F1 | MCC |
|---|---|---|---|
| GCN (ICLR'17) | 0.580±0.013 | 0.566±0.053 | 0.167±0.029 |
| GraphSAGE (NeurIPS'17) | 0.606±0.031 | 0.595±0.055 | 0.214±0.058 |
| GAT (ICLR'18) | 0.604±0.018 | 0.618±0.040 | 0.214±0.038 |
| GIN (ICLR'19) | 0.599±0.028 | 0.623±0.036 | 0.201±0.055 |
| BrainGNN (MedIA'21) | 0.617±0.059 | 0.650±0.075 | 0.245±0.128 |
| IBGNN (MICCAI'22) | 0.604±0.024 | 0.635±0.037 | 0.214±0.049 |
| FBNetGen (MIDL'22) | 0.616±0.009 | 0.609±0.030 | 0.234±0.017 |
| CycGAT (MICCAI'24) | 0.587±0.028 | 0.597±0.062 | 0.183±0.071 |
| BrainOOD (ICLR'25) | 0.607±0.043 | 0.603±0.086 | 0.220±0.091 |
| STAGIN (NeurIPS'21) | 0.552±0.024 | 0.558±0.058 | 0.109±0.050 |
| BioBGT (ICLR'25) | 0.610±0.029 | 0.649±0.027 | 0.227±0.057 |
| DisenGCN (ICML'19) | 0.572±0.049 | 0.590±0.098 | 0.155±0.101 |
| GIB (NeurIPS'20) | 0.629±0.025 | 0.640±0.050 | 0.263±0.053 |
| DGCL (NeurIPS'21) | 0.599±0.018 | 0.636±0.009 | 0.203±0.031 |
| CI-GNN (NN'24) | 0.631±0.012 | 0.629±0.042 | 0.266±0.057 |
| **SMGD (Ours)** | **0.639±0.025** | **0.657±0.042** | **0.286±0.025** |

are further randomly partitioned into training/validation subsets, and we retain the checkpoint with the highest validation accuracy. All hyperparameters are set before the training process and reused across folds.

**Baselines.** We compare against diverse representative methods spanning four categories: (1) ***Classic GNNs***: GCN (Kipf & Welling, 2017), GraphSAGE (Hamilton et al., 2017), GAT (Veličković et al., 2018), GIN (Xu et al., 2019); (2) ***Brain-specific GNNs***: BrainGNN (Li et al., 2021b), IBGNN (Cui et al., 2022), FBNetGen (Kan et al., 2022), CycGAT (Huang et al., 2024), BrainOOD (Xu et al., 2025); (3) ***Transformer-based Methods***: STAGIN (Kim et al., 2021), BioBGT (Peng et al., 2025); (4) ***Disentanglement Methods***: DisenGCN (Ma et al., 2019), GIB (Wu et al., 2020), DGCL (Li et al., 2021a), CI-GNN (Zheng et al., 2024). Since most baselines are originally designed for single-modality graphs and do not specify a multi-modal fusion operator, we adapt them under a unified late-fusion protocol: modality-specific encoder instances are applied to each graph view, and their graph-level representations are concatenated for final prediction. Checkpoints are selected under the same split protocol. For fair comparison, we use a unified hyperparameter grid search (e.g., learning rate, GNN layers) for all baseline models; for model-specific hyperparameters (e.g., the trade-off parameter in GIB), the grid search ranges were centered around the optimal values reported in the original papers; for certain works adapted to the ABIDE/SRPBS datasets (e.g., CI-GNN, BioBGT), we adopted the published configuration parameters.

### 5.2. Performance Comparison

**Within-dataset Classification.** We evaluate SMGD on within-dataset classification using stratified cross-validation: 10-fold on ABIDE-I and 5-fold on SRPBS-SSD. In each

*Table 3.* ABIDE-I performance across fusion/alignment paradigms. $\Delta_{\text{gap}}$ denotes the modality gap (N/A if undefined), whereas $\hat{\Delta}_{\text{gap}}$ is the scale-normalized modality gap.

| Method | $\Delta_{\text{gap}}$ | $\hat{\Delta}_{\text{gap}}$ | ACC | F1 | MCC |
|---|---|---|---|---|---|
| Cross-attention | N/A | N/A | 0.717 | 0.711 | 0.425 |
| Early Fusion | N/A | N/A | 0.602 | 0.594 | 0.201 |
| Mid Fusion | N/A | N/A | 0.713 | 0.704 | 0.437 |
| Late Fusion | 4.785 | 0.899 | 0.704 | 0.691 | 0.424 |
| $\text{SMGD}_{\text{CL}}$ | 1.729 | 0.346 | 0.722 | 0.719 | 0.470 |
| $\text{SMGD}_{\text{MSE}}$ | 2.089 | 0.535 | 0.701 | 0.702 | 0.406 |
| SMGD | 2.203 | 0.907 | **0.748** | **0.725** | **0.507** |

setting, one fold is held out as the test set, and the remaining data are further split into training and validation sets with train/validation ratios of 8:1 (ABIDE-I) and 3:1 (SRPBS-SSD). We report Accuracy (ACC), F1-Score (F1), and Matthews Correlation Coefficient (MCC) as mean ± standard deviation. As shown in Table 1, SMGD achieves strong within-dataset performance, attaining 74.8% ACC on ABIDE-I and 95.4% ACC on SRPBS-SSD.

**Cross-dataset Generalization.** To evaluate robustness, we train models on ABIDE-I with a 4:1 split and directly test on the strictly held-out SRPBS-ASD cohort without fine-tuning. We enforce a rigorous protocol where unified preprocessing is applied, and all data-dependent parameters (e.g., thresholds) are estimated exclusively on the source training set. To ensure fair comparison, all baselines utilize the same dual-modal inputs, with single-modal methods adopting a concatenation-based fusion strategy mirroring SMGD. Results are averaged over 5 runs with different random seeds. As shown in Table 2, SMGD achieves 63.9% ACC in this setting. Notably, SMGD attains this gain without introducing additional dataset-specific adaptation or hand-crafted transfer components, suggesting that a general representation disentanglement principle can yield highly transferable diagnostic features under dataset shift.

### 5.3. Performance and Gap across Paradigms

To evaluate how fusion design and cross-modal consistency regularization affect diagnosis performance, we benchmark six representative paradigms under matched GPS backbones and training protocols, and examine the relation between performance and the modality gap $\Delta_{\text{gap}}$ when it is well-defined. Specifically, we compare the following paradigms: *Cross-attention* (after independent GPS encoding and graph pooling, apply bidirectional multi-head cross-attention between modality tokens with residual connections before fusion), *Early Fusion* (concatenate node features at input via linear projection, then process through a single GPS stack), *Mid Fusion* (separate GPS encoding for $L/2$ layers, concatenate at mid-point through a fusion projection, then

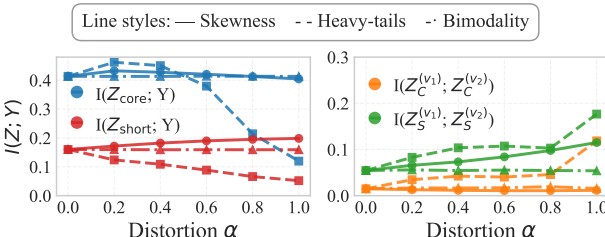

*Figure 3.* Post-hoc stress test of DDH surrogates.

*Table 4.* Ablation analysis on ABIDE-I. *Setting*: removed constraint; *Diagnosis*: optimization objective (core: core components only; short: with shortcut-like components).

| Setting | Diagnosis | $ACC$ | $F1$ | $MCC$ |
|---|---|---|---|---|
| w/o Non-degeneracy | core+short | 0.679 | 0.629 | 0.366 |
| w/o $Z_S$ (Align) | core+short | 0.671 | 0.650 | 0.352 |
| w/o $Z_I$ (Align) | core+short | 0.668 | 0.634 | 0.345 |
| w/o $Z_C$ (Decoup) | core+short | 0.685 | 0.649 | 0.374 |
| w/o $Z_N$ (Decoup) | core+short | 0.699 | 0.672 | 0.402 |
| w/o $L_{short}$ | core | 0.709 | 0.682 | 0.423 |
| w/o SMGD | core | 0.703 | 0.677 | 0.411 |
| Full Model | core+short | **0.748** | **0.725** | **0.507** |

shared GPS layers for remaining depth), *Late Fusion* (two independent GPS stacks with separate projection heads, concatenate graph-pooled embeddings for classification), *SMGD*$_{CL}$ (Late Fusion with an auxiliary InfoNCE loss on pooled modality embeddings), and *SMGD*$_{MSE}$ (replacing SMGD's GW structural alignment with an MSE alignment term). We next analyze the modality gap $\Delta_{gap}$ across different paradigms to assess how it relates to performance. Specifically, we compute $\Delta_{gap}$ as the $\ell_2$ distance between the centroids of the two modality embeddings ($z^{(v_1)}$ and $z^{(v_2)}$). Since this measure is sensitive to representation scale, we also report a scale-normalized variant of $\Delta_{gap}$ by applying $\ell_2$ normalization to each modality's embedding before computing the centroids, ensuring that the results are not influenced by differences in embedding scale.

Table 3 reports diagnostic performance together with both the raw modality gap $\Delta_{gap}$ and its scale-normalized counterpart $\widehat{\Delta}_{gap}$. We find that objectives explicitly minimizing cross-modal mismatch (InfoNCE and MSE) indeed reduce $\Delta_{gap}$, but this reduction does not translate into commensurate performance gains. For example, SMGD$_{CL}$ drives $\Delta_{gap}$ to 1.729 yet still underperforms SMGD by 2.6% in accuracy. This trend is consistent with our theoretical analysis in Appendix B, which predicts that aggressive alignment can attenuate complementary information components. In contrast, SMGD achieves the best performance while maintaining a moderate $\Delta_{gap}$, indicating that preserving complementary information can be more beneficial than eliminating the modality gap.

### 5.4. Stress Testing for DDH Surrogates

To examine how sensitive our DDH-motivated surrogates are to violations of approximate Gaussianity, we conduct a post-hoc stress test on ABIDE-I. Starting from a trained SMGD checkpoint (parameters frozen), we extract the learned subspace representations and apply three controlled non-Gaussian distortions with strength $\alpha \in [0, 1]$: (i) *Skewness* via element-wise power transform $X' = \text{sign}(X_{norm})|X_{norm}|^{1+\alpha}$; (ii) *Heavy-tails* by mixing observations with $t$-distributed samples where degrees of freedom drop from 30 to 2 as $\alpha \to 1$; and (iii) *Bimodality*, induced by shifting two randomly split sample subsets by $\pm 4\alpha$. To

track subspace-wise dependence efficiently, we employ a fast correlation-based dependence proxy inspired by Gaussian canonical correlation: $I(Z; Y) \approx -\frac{1}{2}\log(1 - \rho^2)$, where $\rho^2$ is the maximum eigenvalue of the generalized correlation matrix $M = \Sigma_{ZZ}^{-1}\Sigma_{ZY}\Sigma_{YY}^{-1}\Sigma_{YZ}$.

As shown in Figure 3, the dependence remains relatively stable under Skewness and Heavy-tails but exhibits noticeable drift under strongly bimodal distortions, delineating the boundary where covariance-based surrogates may weaken. Dataset-level DDH validation is deferred to Appendix E.

### 5.5. Ablation Analysis

To investigate the contribution of each component, we conduct a systematic ablation study on ABIDE-I as shown in Table 4. We first observe that specific geometric constraints provide complementary gains. The removal of the non-degeneracy constraint leads to noticeable degradation (0.679), while disabling the alignment for shared diagnostic features (w/o $Z_S$) causes a substantial drop to 0.671, identifying cross-modal consensus as a foundational driver of performance. Beyond these foundations, the omission of the shortcut regularizer (w/o $L_{short}$) reduces accuracy to 0.709, validating the necessity of actively suppressing shortcut-related nuisances. Furthermore, the comparison between the Full Model and the w/o SMGD setting, which reduces to a basic backbone encoder with fusion, reveals a clear performance gap. These results confirm the necessity of strategic factorization for accurate diagnosis.

### 5.6. Interpretability Analysis

We conduct ROI-level attribution on ABIDE-I to assess whether SMGD relies on neurobiologically plausible signals and to probe how evidence is allocated across modalities and disentangled components. We use Integrated Gradients to obtain subject-wise ROI importance scores, aggregate them to cohort-level maps, and visualize the Top-$K$ ROIs on the Brainnetome-246 atlas (Top-16 per modality; Top-10 per information component). Modality-level attribution targets the ASD prediction logit, whereas component-level

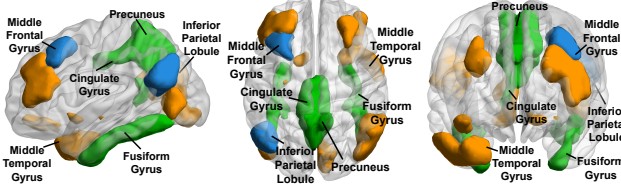

*(a)* Sagittal view    *(b)* Axial view    *(c)* Coronal view

*Figure 4.* Cortical surface attributions of different modalities. Regions specific to FC and SS are colored in green and orange, respectively, while their intersection is highlighted in blue.

attribution targets each subspace's activation energy. Full implementation details are provided in Appendix G.

As shown in Figure 4, FC attributions concentrate on fusiform regions and Default Mode Network (DMN) hubs including the precuneus and posterior cingulate, consistent with ASD findings on default-mode and social-perceptual circuitry (Oblak et al., 2011; Cheng et al., 2015; Padmanabhan et al., 2017). SS emphasizes Middle Frontal Gyrus (MFG), Middle Temporal Gyrus (MTG) and Superior Temporal Gyrus (STG), consistent with reported morphometric alterations in fronto-temporal association cortex and atypical social-perceptual processing in ASD (Ecker et al., 2013; Yang et al., 2016). Notably, FC and SS show overlapping attribution on several ASD-relevant hubs, including MFG and Inferior Parietal Lobule (IPL), suggesting a shared diagnostic substrate, with modality-specific ROIs providing additional complementary evidence (Nomi & Uddin, 2015). The ROIs that most differentiate the four components are further reported in Appendix G.

## 6. Limitations and Further Work

While SMGD provides a structured formulation for separating diagnostic evidence from shortcut-related information, its current design should be understood within the scope of dual-modal neuroimaging diagnosis. Extending this framework to scenarios involving three or more modalities remains challenging, as higher-order Partial Information Decomposition introduces a rapidly increasing number of information atoms and lacks a universally accepted closed-form solution. Another limitation lies in the use of DDH-based geometric surrogates, where covariance statistics are adopted to approximate information-theoretic constraints. This approximation is empirically supported in our experiments, but may become less reliable when representations exhibit heavy-tailed behavior, strong nonlinear dependencies, severe outliers, or highly non-unimodal structures. These observations suggest that future work could incorporate more robust dependence measures, such as kernel-based or neural estimators, to relax the Gaussian approximation.

Future work may advance SMGD in two methodologi-

cal directions. First, the current dual-modal formulation could be extended to scalable higher-order multi-modal disentanglement, enabling more faithful characterization of shared, complementary, and shortcut-related factors across functional, structural, diffusion, genetic, or clinical variables. Second, covariance-based DDH surrogates could be replaced or complemented with more robust dependence measures, such as kernel-based criteria or neural mutual-information estimators, to relax the Gaussian approximation and improve reliability under non-Gaussian or nonlinear representation distributions.

## 7. Conclusion

We presented SMGD, which recasts multi-modal neuroimaging graph diagnosis as structured information learning and unifies alignment and fusion within a single factorized representation space. SMGD decomposes each modality embedding into shared and complementary diagnostic factors, together with incidental agreement and modality-specific nuisance factors, and enforces their designated roles via mini-batch geometric surrogate regularizers derived under a mild distributional assumption. Experiments on ABIDE-I and SRPBS show consistent gains in diagnosis and cross-dataset generalization, and ablations verify the benefit of the proposed factorization and geometry-driven constraints. These results suggest that explicitly structuring the multi-modal representation space is a promising direction for building generalizable neuroimaging diagnosis models.

## Acknowledgments

This work was supported by Noncommunicable Chronic Diseases-National Science and Technology Major Project (2024ZD0527800) and the National Natural Science Foundation of China with grant numbers (32541016, U25A20540, 62436005, 62473303).

## Code and Data Availability

Code and preprocessed data for both datasets are provided at:https://github.com/greedisgood1000/SMGD. Raw data are available upon reasonable request.

## Ethical Statement

There are no ethical issues.

## Impact Statement

This work aims to improve computer-aided diagnosis of psychiatric disorders. While we believe the potential clinical benefits are positive, any practical deployment would require rigorous clinical validation.

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

# A. PID and SMGD: A Conceptual Comparison

Partial Information Decomposition (PID) studies how multiple sources contribute to a target variable. For two predictors $Z^{(v_1)}, Z^{(v_2)}$ and target $Y$, PID decomposes the total task-relevant information into four non-negative atoms (Bertschinger et al., 2014; Liang et al., 2023a;b; Dewan et al., 2024):

$$I(Z^{(v_1)}, Z^{(v_2)}; Y) = I_\cap + U_1 + U_2 + I_{\text{syn}}, \tag{17}$$

where $I_\cap$ denotes *redundancy* (information both sources can provide about $Y$), $U_1, U_2$ denote *uniqueness* (source-specific information about $Y$), and $I_{\text{syn}}$ denotes *synergy* (information about $Y$ only available when sources are combined). These atoms satisfy the standard consistency identities:

$$I_\cap + U_1 = I(Z^{(v_1)}; Y), \quad I_\cap + U_2 = I(Z^{(v_2)}; Y), \tag{18}$$

$$U_1 + I_{\text{syn}} = I(Z^{(v_1)}; Y \mid Z^{(v_2)}), \quad U_2 + I_{\text{syn}} = I(Z^{(v_2)}; Y \mid Z^{(v_1)}). \tag{19}$$

PID is therefore task-centric: it classifies information by how it contributes to predicting $Y$, and it does so by explicitly modeling multi-source interaction (redundancy vs. synergy vs. uniqueness). In contrast, SMGD is representation-centric: it asks how a multi-modal latent space should be factorized so that (i) task-relevant signals are preserved but (ii) confounds and modality-specific nuisance are isolated, while still permitting cross-modal integration under the modality gap. Concretely, SMGD posits that each modality representation consists of four subspaces,

$$Z^{(v)} = [Z_S^{(v)}, Z_C^{(v)}, Z_I^{(v)}, Z_N^{(v)}], \tag{20}$$

characterized by the joint disentanglement of cross-modal consistency (shared vs. specific) and diagnostic utility (core vs. shortcut).

Under this lens, PID and SMGD can be aligned at the level of $Y$-relevant contributions: redundancy and synergy are intrinsically cross-source phenomena about $Y$, so their aggregate naturally corresponds to a shared-relevant notion, whereas uniqueness corresponds to specific-relevant information:

$$I(Z^{(v_1)}, Z^{(v_2)}; Y) = \underbrace{(I_\cap + I_{\text{syn}})}_{\text{shared-relevant}} + \underbrace{(U_1 + U_2)}_{\text{specific-relevant}}. \tag{21}$$

Crucially, the Venn-style intersection between $Z^{(v_1)}$ and $Z^{(v_2)}$ reflects the total cross-modal dependence $I(Z^{(v_1)}; Z^{(v_2)})$, which does not coincide with $I_\cap$ (or $I_\cap + I_{\text{syn}}$) because it also captures shared spurious correlations. Algebraically, the cross-modal mutual information decomposes as

$$I(Z^{(v_1)}; Z^{(v_2)}) = I(Z^{(v_1)}; Z^{(v_2)}; Y) + I(Z^{(v_1)}; Z^{(v_2)} \mid Y), \tag{22}$$

where $I(Z^{(v_1)}; Z^{(v_2)} \mid Y)$ may capture incidental agreement not aligned with the prediction mechanism $Z \to Y$. SMGD treats this term as a first-class object by explicitly splitting the shared intersection into $Z_S$ (shared-relevant) and $Z_I$ (shared-shortcut), thereby reducing spurious cross-modal correlations under distribution shift.

Accordingly, we can view the correspondence between the two formalisms as a semantic alignment rather than a strict derivation:

$$\begin{aligned} Z_S &\rightsquigarrow I_\cap + I_{\text{syn}}, \qquad Z_C^{(v)} \rightsquigarrow U_v, \\ Z_I &\rightsquigarrow \text{label-neutral shared agreement}, \\ Z_N^{(v)} &\rightsquigarrow \text{label-neutral modality-private residual variation}. \end{aligned} \tag{23}$$

Here, $Z_S$ captures redundant and synergistic cross-modal diagnostic evidence, whereas $Z_C^{(v)}$ captures modality-specific diagnostic evidence. $Z_I$ and $Z_N^{(v)}$ are not PID atoms, but SMGD-specific shortcut-related components. In particular, $Z_N^{(v)}$ is defined as a label-neutral, modality-private, and non-degenerate residual subspace satisfying:

$$I(Z_N^{(v)}; Y) \le \epsilon_y, \qquad I(Z_N^{(v_1)}; Z_N^{(v_2)}) \le \epsilon_m, \qquad H(Z_N^{(v)}) \ge h_0. \tag{24}$$

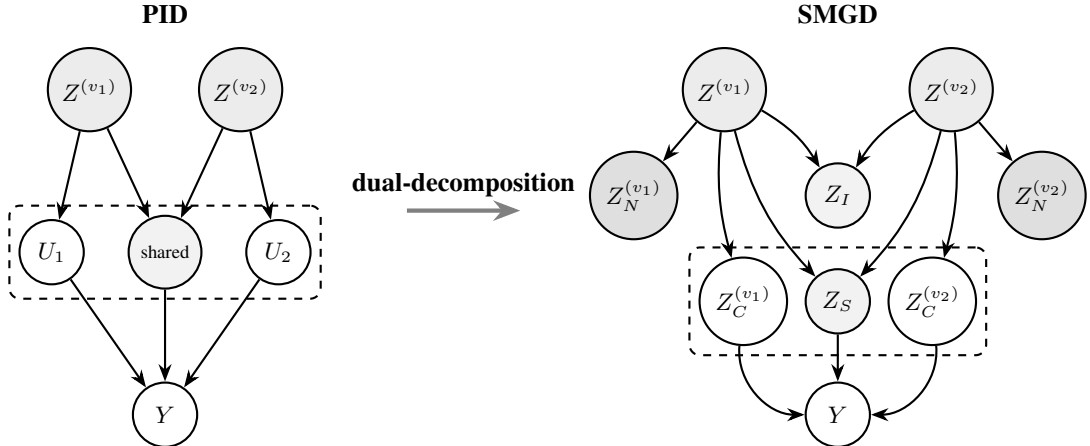

*Figure 5.* From PID to SMGD. **Left**: PID decomposes modality information into shared and unique components. **Right**: SMGD introduces a diagnostic dimension, yielding a $2 \times 2$ structure. Upper row: shortcut-like nuisances ($Z_N^{(v)}, Z_I$); lower row: diagnostic core ($Z_C^{(v)}, Z_S$). The dashed box indicates the task-relevant subspaces employed to predict $Y$.

## B. Theoretical Analysis of the Modality Gap in Alignment Paradigms

This section investigates three alignment paradigms: Mean Squared Error (MSE), Information Noise-Contrastive Estimation (InfoNCE) (Oord et al., 2018), and our GW-based approach. We first elucidate the theoretical reasons why MSE and InfoNCE are susceptible to the modality gap. Subsequently, we show how SMGD is theoretically grounded to achieve semantic alignment without eliminating this gap.

### B.1. Definition and Geometric Origin of the Modality Gap

The modality gap is a persistent geometric phenomenon in multi-modal representation learning, especially in contrastive alignment settings, commonly manifested as a non-vanishing centroid mismatch between modality-specific embedding clouds.

**Definition B.1** (Modality gap: population and empirical forms). Let $Z^{(v)}$ denote the representation of modality $v \in \{v_1, v_2\}$, and let $\mu^{(v)} = \mathbb{E}[Z^{(v)}]$ denote its population centroid. The population modality gap is defined as $\Delta_{\text{gap}} = \left\| \mu^{(v_1)} - \mu^{(v_2)} \right\|_2$. Given $n$ paired samples, its empirical estimate is

$$\widehat{\Delta}_{\text{gap}} = \left\| \hat{\mu}^{(v_1)} - \hat{\mu}^{(v_2)} \right\|_2, \quad \hat{\mu}^{(v)} = \frac{1}{n} \sum_{i=1}^{n} z_i^{(v)}. \tag{25}$$

A key reason why $\Delta_{\text{gap}}$ may remain positive even after extensive training is the cone effect in deep neural networks. Under commonly studied random-weight settings, successive ReLU layers can shrink the angular spread of activations. For example, for a feedforward layer with random weights $W$ and bias $b$, the cosine similarity may satisfy

$$\cos\left(\sigma(Wu + b), \sigma(Wv + b)\right) > \cos(u, v), \tag{26}$$

with high probability under suitable conditions, where $\sigma$ denotes ReLU. Stacking such layers can concentrate embeddings into narrow, encoder-specific cones even at initialization. In multi-modal settings, separate encoders may therefore start from distinct conical regions and remain partially separated, making complete cross-modal overlap difficult to achieve through optimization alone.

### B.2. Effect of the Modality Gap on Contrastive Alignment

Contrastive learning maximizes cross-modal agreement via objectives such as InfoNCE. However, we show that its optimization mechanism inherently preserves a residual modality gap and, in doing so, suppresses complementary diagnostic information.

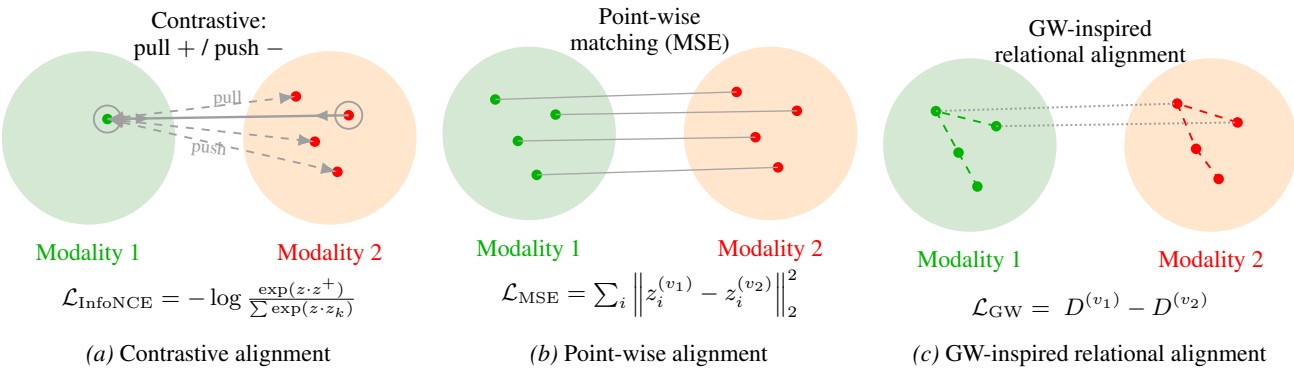

*(a)* Contrastive alignment        *(b)* Point-wise alignment        *(c)* GW-inspired relational alignment

*Figure 6.* Schematic comparison of three cross-modal alignment strategies: contrastive (pull positives and push negatives), point-wise matching, and GW-inspired relational alignment that preserves intra-modal geometry.

Consider the InfoNCE objective that maximizes cosine similarity between positive pairs $(Z^{(v_1)}, Z^{(v_2)})$. Because the representation is concatenated over disjoint blocks, the inner product decomposes as

$$\cos(Z^{(v_1)}, Z^{(v_2)}) = \frac{\sum_{\mathcal{A} \in \{C,S,I,N\}} \langle Z_{\mathcal{A}}^{(v_1)}, Z_{\mathcal{A}}^{(v_2)} \rangle}{\|Z^{(v_1)}\|_2 \cdot \|Z^{(v_2)}\|_2}. \tag{27}$$

For the cross-modally shared components, the SMGD design encourages positive cross-modal correspondence or relational consistency between $Z_S^{(v1)}$ and $Z_S^{(v2)}$, and between $Z_I^{(v1)}$ and $Z_I^{(v2)}$. Thus, their paired inner products tend to contribute positively to the numerator. However, for the modality-specific components ($Z_C$ and $Z_N$), the SMGD framework requires cross-modal independence: $I(Z_C^{(v_1)}; Z_C^{(v_2)}) = 0$ and $I(Z_N^{(v_1)}; Z_N^{(v_2)}) = 0$. Under approximate Gaussianity (DDH), independence implies zero correlation, hence these cross-modal inner products vanish in expectation:

$$\mathbb{E}\big[\langle Z_C^{(v_1)}, Z_C^{(v_2)} \rangle\big] = 0, \quad \mathbb{E}\big[\langle Z_N^{(v_1)}, Z_N^{(v_2)} \rangle\big] = 0. \tag{28}$$

Consequently, only the shared components contribute stably to the numerator:

$$\cos(Z^{(v_1)}, Z^{(v_2)}) \approx \frac{\|Z_S\|_2^2 + \|Z_I\|_2^2}{\|Z^{(v_1)}\|_2 \cdot \|Z^{(v_2)}\|_2}. \tag{29}$$

By the independent decomposition, the denominator expands as:

$$\|Z^{(v)}\|_2^2 = \|Z_S\|_2^2 + \|Z_I\|_2^2 + \|Z_C^{(v)}\|_2^2 + \|Z_N^{(v)}\|_2^2. \tag{30}$$

Although $Z_C$ and $Z_N$ do not contribute to the numerator in Eq. (29), they inflate the denominator through their squared norms. This creates a dilution effect: increasing $\|Z_C\|_2$ or $\|Z_N\|_2$ enlarges the denominator while leaving the numerator unchanged, thereby reducing the cosine similarity.

To maximize the contrastive objective, the positive-pair alignment pressure incentivizes the optimizer to reduce the energy of modality-specific components. Critically, the objective cannot distinguish between $Z_N$ and $Z_C$. Consequently, both are penalized equally as sources of dilution.

As a result, the representation learned by contrastive alignment is biased toward the shared components. Under the approximation that modality-specific components are suppressed, $Z_{\text{contrastive}}$ is primarily a function of $[Z_S, Z_I]$. By the data processing inequality,

$$I(Z_{\text{contrastive}}; Y) \leq I([Z_S, Z_I]; Y) = I(Z_S; Y) + I(Z_I; Y \mid Z_S) \approx I(Z_S; Y), \tag{31}$$

where the approximation follows from the SMGD requirement that $Z_I$ carries negligible diagnostic information.

### B.3. MSE Alignment Causes Information Loss When Bridging the Modality Gap

Given the persistent modality gap described above, a natural remedy is to minimize point-wise MSE between cross-modal representations. We analyze the consequences of this strategy under the SMGD framework, showing that eliminating the modality gap via MSE inevitably suppresses complementary diagnostic information.

Applying Jensen's inequality to the convex function $\| \cdot \|_2^2$ yields a lower bound:

$$\mathcal{L}_{\text{MSE}} = \mathbb{E}\big[\|Z^{(v_1)} - Z^{(v_2)}\|_2^2\big] \geq \left\|\mathbb{E}[Z^{(v_1)}] - \mathbb{E}[Z^{(v_2)}]\right\|_2^2 = \Delta_{\text{gap}}^2. \tag{32}$$

This inequality implies that MSE alignment closes the modality gap through a stronger point-wise constraint: $\mathcal{L}_{\text{MSE}} \to 0 \Rightarrow \Delta_{\text{gap}} \to 0$. Because the representation is concatenated over disjoint blocks, the MSE loss decomposes as:

$$\mathcal{L}_{\text{MSE}} = \sum_{\mathcal{A} \in \{C,S,I,N\}} \mathbb{E}\big[\|Z_{\mathcal{A}}^{(v_1)} - Z_{\mathcal{A}}^{(v_2)}\|_2^2\big]. \tag{33}$$

Since each term is non-negative, driving $\mathcal{L}_{\text{MSE}} \to 0$ forces every subspace term to vanish. For the shared subspaces ($Z_S$ and $Z_I$), this alignment is semantically appropriate. However, for the modality-specific complementary subspace $Z_C$, this leads to:

$$\mathbb{E}\left[\|Z_C^{(v1)} - Z_C^{(v2)}\|_2^2\right] \to 0. \tag{34}$$

In the zero-loss limit, since $\mathbb{E}\|X\|_2^2 = 0$ implies $X = 0$ almost surely for any random variable $X$, we obtain

$$Z_C^{(v1)} = Z_C^{(v2)} \quad \text{a.s.,}$$

without requiring any distributional assumption. To establish the information-theoretic consequence of Eq. (34), we invoke the following lemma.

**Lemma B.2** (Deterministic Collapse under Equality and Conditional Independence). *Let $U$, $V$, and $W$ be random variables. If $U = V$ almost surely and $I(U;V \mid W) = 0$, then there exists a measurable function $g$ such that $U = g(W)$ almost surely.*

*Proof.* The condition $I(U;V \mid W) = 0$ implies conditional independence between $U$ and $V$ given $W$. Since $U = V$ almost surely, for almost every value of $W$, the conditional joint distribution of $(U,V)$ is supported on the diagonal set $\{(u,v) : u = v\}$. A product distribution supported on this diagonal must have a degenerate marginal. Hence, the conditional distribution of $U$ given $W$ is almost surely a point mass, which implies that $U = g(W)$ almost surely for some measurable function $g$. $\square$

For the purpose of analyzing the collapse induced by full MSE alignment, we consider an idealized SMGD factorization in which the complementary subspaces are modality-private after conditioning on the shared diagnostic component:

$$I(Z_C^{(v_1)}; Z_C^{(v_2)} \mid Z_S) = 0. \tag{35}$$

Combining Eq. (34) and Eq. (35) with Lemma B.2, we obtain

$$Z_C^{(v_1)} = g(Z_S) \quad \text{a.s.} \tag{36}$$

for some measurable function $g$. Equivalently, the concatenated complementary representation $Z_C = [Z_C^{(v_1)}, Z_C^{(v_2)}]$ becomes a deterministic function of $Z_S$. Therefore, it cannot provide any additional label-relevant information beyond the shared component:

$$I(Z_C; Y \mid Z_S) = 0. \tag{37}$$

Consequently, the total diagnostic information in the concatenated representation collapses:

$$I([Z_S, Z_C]; Y) = I(Z_S; Y) + I(Z_C; Y \mid Z_S) \xrightarrow{\mathcal{L}_{\text{MSE}} \to 0} I(Z_S; Y). \tag{38}$$

The lost term $I(Z_C; Y \mid Z_S)$ is precisely the complementary diagnostic information. This quantity is strictly positive whenever different modalities provide non-redundant diagnostic signals, which is the typical case in multi-modal neuroimaging where functional and structural data capture distinct aspects of pathology.

### B.4. SMGD Achieves Semantic Alignment Without Eliminating the Modality Gap

We now demonstrate that SMGD does not claim to eliminate the modality gap. Instead, it achieves cross-modal semantic alignment and avoids explicit information loss while tolerating the persistent centroid mismatch across modalities.

Recall that the modality gap is defined as $\Delta_{\text{gap}} = \|\mathbb{E}[Z^{(v_1)}] - \mathbb{E}[Z^{(v_2)}]\|_2$. Under the SMGD decomposition $Z^{(v)} = [Z_S^{(v)}, Z_C^{(v)}, Z_I^{(v)}, Z_N^{(v)}]$, the squared gap decomposes as:

$$\Delta_{\text{gap}}^2 = \|\Delta\mu_S\|^2 + \|\Delta\mu_C\|^2 + \|\Delta\mu_I\|^2 + \|\Delta\mu_N\|^2, \tag{39}$$

where $\Delta\mu_{\mathcal{A}} = \mathbb{E}[Z_{\mathcal{A}}^{(v_1)}] - \mathbb{E}[Z_{\mathcal{A}}^{(v_2)}]$ denotes the centroid mismatch for subspace $\mathcal{A}$.

We now analyze how SMGD's loss functions constrain each term. For shared subspaces $(Z_S, Z_I)$, the relational alignment loss is:

$$\mathcal{L}_{\text{align}}(Z_{\mathcal{A}}) = \frac{1}{B^2} \sum_{i,j} \left( \widetilde{D}_{\mathcal{A},ij}^{(v_1)} - \widetilde{D}_{\mathcal{A},ij}^{(v_2)} \right)^2, \tag{40}$$

where

$$D_{\mathcal{A},ij}^{(v)} = \|z_{\mathcal{A},i}^{(v)} - z_{\mathcal{A},j}^{(v)}\|_2, \qquad \widetilde{D}_{\mathcal{A}}^{(v)} = \frac{D_{\mathcal{A}}^{(v)}}{\|D_{\mathcal{A}}^{(v)}\|_F + \epsilon_d}.$$

This loss is invariant to isometric transformation $z_{\mathcal{A},i}^{(v_2)} = z_{\mathcal{A},i}^{(v_1)} \cdot R + t$, where $R \in O(d_Z)$ and $t \in \mathbb{R}^{d_Z}$. Since pairwise distances are preserved under translation, the relational alignment $\mathcal{L}_{\text{align}} = 0$ does not require $\Delta\mu_S = 0$ or $\Delta\mu_I = 0$.

For specific subspaces $(Z_C, Z_N)$, the cross-modal independence loss is:

$$\mathcal{L}_{\text{decoup}}(Z_{\mathcal{B}}) = \|\text{Cov}(Z_{\mathcal{B}}^{(v_1)}, Z_{\mathcal{B}}^{(v_2)})\|_F^2, \quad \text{where} \quad \text{Cov}(Z_{\mathcal{B}}^{(v_1)}, Z_{\mathcal{B}}^{(v_2)}) = \frac{1}{B-1} \bar{Z}_{\mathcal{B}}^{(v_1)\top} \bar{Z}_{\mathcal{B}}^{(v_2)}, \tag{41}$$

where $\bar{Z}$ denotes the centered representation. Since covariance operates on centered variables, this loss constrains only linear correlations and is completely agnostic to the centroids $\mu^{(v_1)}$ and $\mu^{(v_2)}$ themselves.

Hence, non-zero centroid mismatches can persist without conflicting with the intended subspace structure. For specific subspaces, the centroid mismatch $\|\Delta\mu_C\|^2 + \|\Delta\mu_N\|^2$ precisely reflects the modality-specific nature of these representations. Rather than being a nuisance to eliminate, this mismatch is an expected consequence of preserving complementary information. As established in Sections B.2 and B.3, aggressive contrastive or point-wise alignment can reduce cross-modal mismatch while attenuating modality-specific complementary information:

$$I(Z_{\text{MSE}}; Y), \ I(Z_{\text{contrastive}}; Y) \ \xrightarrow{\text{alignment}} \ I(Z_S; Y). \tag{42}$$

In contrast, SMGD allows the complementary diagnostic information to be retained by tolerating the gap:

$$I(Z_{\text{SMGD}}; Y) \to I(Z_S; Y) + I(Z_C; Y \mid Z_S). \tag{43}$$

The information gain $\Delta I_{\text{SMGD}} = I(Z_C; Y \mid Z_S) \geq 0$ can be retained because SMGD does not force the modality gap to vanish.

## C. Model Architecture

The proposed geometric regularization method is integrated within a GNN pipeline consisting of two modality-specific *General, Powerful, Scalable (GPS) Graph Transformer* encoders (Rampášek et al., 2022), a hierarchical disentanglement head for quadruple representation decomposition, and a classification head for final prediction. For clarity, we describe the propagation process on a single graph; the extension to batch processing is straightforward. Let $h_i^l \in \mathbb{R}^{d_H}$ denote the feature vector of node $i$ at layer $l$, and $H^l = [h_1^l; \ldots; h_N^l]^\top \in \mathbb{R}^{N \times d_H}$ denote the stacked node feature matrix.

**Hybrid Graph Encoding.**   Starting from $H^{(v),0} = \text{GELU}(\text{BN}(X^{(v)} W_0))$, where BN denotes Batch Normalization, the encoder updates representations through stacked GPS layers, each comprising a local and a global branch.

The local branch employs a residual gated Graph ConvNet for neighborhood aggregation. Let $\mathcal{N}_{\text{in}}(i)$ denote the set of source nodes $j$ such that there exists a directed edge $j \to i$, then:

$$h_{i\text{-local}}^{(v),l+1} = \text{ReLU}\Big(h_i^{(v),l} + \text{BN}\Big(\sum_{j \in \mathcal{N}_{\text{in}}(i)} \eta_{j \to i} \odot (W_h h_j^{(v),l} + W_e\, e_{j,i})\Big)\Big), \tag{44}$$

where $\odot$ denotes the Hadamard product and $e_{j,i}$ is the edge weight associated with $j \to i$. The gating coefficient modulates message importance:

$$\eta_{j \to i} = \sigma\Big(W_g[\, h_j^{(v),l} \,\|\, h_i^{(v),l} \,\|\, e_{j,i}\,]\Big) \in \mathbb{R}^{d_H}, \tag{45}$$

where $\sigma(\cdot)$ is the sigmoid function and $\|\cdot\|$ denotes concatenation.

The global branch applies Multi-head self-attention (MHSA) with Layer Normalization (LN) to capture long-range dependencies:

$$H_{\text{global}}^{(v),l+1} = \text{LN}(H^{(v),l} + \text{MHSA}(H^{(v),l})). \tag{46}$$

Given input $H^{(v),l} \in \mathbb{R}^{N \times d_H}$, MHSA computes queries, keys, and values as $Q = H^{(v),l}W_Q$, $K = H^{(v),l}W_K$, $V = H^{(v),l}W_V$ with $W_Q, W_K, W_V \in \mathbb{R}^{d_H \times d_H}$, and outputs $\text{MHSA}(H) = [\text{head}_1, \ldots, \text{head}_h]W_O$, where $\text{head}_i = \text{Softmax}(Q_i K_i^\top / \sqrt{d_k})V_i$, $d_k = d_H/h$, and $h = 4$ is the number of attention heads. A key padding mask is applied to handle variable-sized graphs within a batch.

Each GPS layer maintains its own learnable scalar $\alpha^{(l)}$, constrained to $(0, 1)$ via sigmoid, to balance the two branches:

$$H^{(v),l+1} = \text{LN}\Big(\sigma(\alpha^{(l)})H_{\text{local}}^{(v),l+1} + (1 - \sigma(\alpha^{(l)}))H_{\text{global}}^{(v),l+1}\Big). \tag{47}$$

**Hierarchical Disentanglement Head.** We adopt a coarse-to-fine strategy to decompose $H^{(v)}$ into four factorized subspaces driven by dual optimization objectives.

In Stage 1, a two-layer MLP with LayerNorm and GELU projects $H^{(v)}$ to double the feature dimension, then splits it equally to separate the diagnostic core from shortcut-like nuisances:

$$[H_{\text{core}}^{(v)}, H_{\text{short}}^{(v)}] = \text{Split}(f_{\text{diagnosis}}(H^{(v)})), \tag{48}$$

where $\text{Split}(\cdot)$ divides the feature dimension equally into two parts. Stage 2 then factorizes each component from the perspective of cross-modal consistency using two parallel networks with the same architecture but separate parameters:

$$[H_S^{(v)}, H_C^{(v)}] = \text{Split}(f_{\text{modal}}(H_{\text{core}}^{(v)})), \tag{49}$$

$$[H_I^{(v)}, H_N^{(v)}] = \text{Split}(f_{\text{modal}'}(H_{\text{short}}^{(v)})). \tag{50}$$

Each subspace representation then passes through an independent projection head $\phi_{\mathcal{A}}$ followed by Global Average Pooling (GAP):

$$Z_{\mathcal{A}}^{(v)} = \text{GAP}(\phi_{\mathcal{A}}(H_{\mathcal{A}}^{(v)})), \quad \mathcal{A} \in \{S, C, I, N\}. \tag{51}$$

**Classification Head.** Diagnostically relevant representations from both modalities are concatenated and fed into a 3-layer MLP classifier:

$$\hat{y} = \text{MLP}([Z_S^{(v_1)}, Z_S^{(v_2)}, Z_C^{(v_1)}, Z_C^{(v_2)}]). \tag{52}$$

The shortcut-related representations are excluded from classification but regularized via $\mathcal{L}_{\text{short}}$ constraints to reduce their label-predictive dependence.

# D. Dataset and Preprocessing Details

### D.1. Psychiatric Dataset Details

**ABIDE-I** (Autism Brain Imaging Data Exchange I) is a grassroots consortium aggregating resting-state functional magnetic resonance imaging (rs-fMRI) and structural MRI data from 17 international imaging sites across North America and Europe. The full dataset comprises 1,112 subjects, with ages ranging from 7 to 64 years. In this study, we utilize 1,071 subjects after quality control, including 516 individuals with ASD and 555 typically developing (TD) controls. Data were acquired using scanners from multiple manufacturers (Siemens, Philips, GE) with heterogeneous acquisition parameters across sites, including varying Repetition Times (TR: 1.5–3.0 s), spatial resolutions, and scan durations.

**SRPBS** (Strategic Research Program for Brain Sciences) is a multi-site, multi-disorder neuroimaging database compiled by the DecNef Consortium in Japan, comprising rs-fMRI and structural MRI from 993 patients with various psychiatric disorders and 1,421 healthy controls (HC) collected at 8 sites using 14 scanners (Siemens, Philips, GE). Unlike ABIDE-I, SRPBS employs a unified imaging protocol across sites (10-minute rs-fMRI acquisition, TR=2.5 s, 240 volumes) to minimize inter-site variability. The dataset covers multiple psychiatric conditions including ASD, SSD, Major Depressive Disorder (MDD), Obsessive-Compulsive Disorder (OCD), and Bipolar Disorder, along with neurological conditions such as chronic pain and stroke. From SRPBS, we select the two major disorder cohorts, which consist of 125 patients with ASD and 147 patients with schizophrenia spectrum disorder, along with demographically matched controls to form the SRPBS-ASD and SRPBS-SSD sub-datasets. For our experiments, we focus on the SSD classification and ASD cross-dataset generalization task to evaluate performance. Detailed demographic characteristics of the participants are presented in Table 5.

*Table 5.* Demographic characteristics of participants across datasets.

| Characteristic | | ABIDE-I | SRPBS-ASD | SRPBS-SSD |
|---|---|---|---|---|
| Sample Size | Total | 1,071 | 250 | 294 |
| | Sites | 17 | 2 | 4 |
| Diagnosis | Patient (ASD/SSD) | 516 | 125 | 147 |
| | Control (TD/HC) | 555 | 125 | 147 |
| Sex | Male | 914 | 218 | 172 |
| | Female | 157 | 32 | 122 |
| Age (years) | Mean | 16.8 | 32.4 | 38.0 |

### D.2. Brain Graph Construction

**Preprocessing.** All data were preprocessed using DPABI [1] with a unified pipeline including realignment, normalization, and nuisance regression. Specifically, preprocessing steps include: (1) discarding the first 5 volumes for signal stabilization; (2) slice timing correction; (3) head motion correction with 24-parameter nuisance regression; (4) spatial normalization to Montreal Neurological Institute (MNI) space; (5) nuisance signal regression (white matter, cerebrospinal fluid (CSF), and global signals); and (6) bandpass filtering (0.01–0.1 Hz).

We utilize the human brainnetome atlas (BNA) to parcellate the brain into 246 regions of interest (ROIs). From both rs-fMRI and T1-weighted structural MRI, we construct functional connectivity graphs and structural similarity graphs for each participant.

**Functional Connectivity Graph.** Functional graphs are built using Fisher $z$-transformed Pearson correlations of fMRI time series. Mean Blood-Oxygen-Level-Dependent (BOLD) time series are extracted from each of the 246 ROIs, and functional connectivity matrices are computed as the Fisher $z$-transformed Pearson correlation coefficients between all ROI pairs, yielding a $246 \times 246$ symmetric matrix for each participant.

**Structural Similarity Graph.** Structural similarity graphs are constructed by measuring morphological dissimilarity between gray matter (GM) distributions and then converting it into a similarity score. Structural images are first segmented into GM probability maps using voxel-based morphometry (VBM). For each pair of brain regions $(u, v)$, we compute the 1-Wasserstein distance (Earth Mover's Distance) between their GM volume distributions:

$$D(u,v) = W_1(p_u, p_v) = \inf_{\pi \in \Gamma(p_u, p_v)} \int_{\mathbb{R} \times \mathbb{R}} |x - y| \, d\pi(x, y), \tag{53}$$

where $p_u$ and $p_v$ denote the empirical GM distributions extracted from ROIs $u$ and $v$, and $\Gamma(p_u, p_v)$ is the set of couplings with marginals $p_u$ and $p_v$. Since smaller $D(u,v)$ indicates higher morphological similarity, we convert distances to similarities via a monotonic kernel:

$$S(u,v) = \frac{1}{1+D(u,v)}. \tag{54}$$

---

[1] http://www.rfmri.org/dpabi

# E. Validation of Gaussianity

The theoretical foundation of our geometric regularization framework rests on the *Data Distribution Hypothesis (DDH)*, which posits that the outputs of deterministic neural networks can be approximately Gaussian under specific conditions. This assumption is crucial because it justifies using second-order covariance statistics as a principled proxy for enforcing statistical independence. To empirically examine the Gaussian approximation invoked by DDH, we conduct an empirical assessment of the learned representations from the ABIDE-I test set. Specifically, we compute the mean absolute skewness and excess kurtosis for each component, both falling well within acceptable ranges for approximate normality as shown in Table 6. These results suggest that the learned subspace representations are broadly consistent with the Gaussian approximation invoked by DDH.

*Table 6.* Quantitative Gaussianity assessment (Skewness and Kurtosis) of learned representations.

| **Metric** | $Z_S^{(v_1)}$ | $Z_S^{(v_2)}$ | $Z_C^{(v_1)}$ | $Z_C^{(v_2)}$ | $Z_I^{(v_1)}$ | $Z_I^{(v_2)}$ | $Z_N^{(v_1)}$ | $Z_N^{(v_2)}$ |
|---|---|---|---|---|---|---|---|---|
| Skewness | 0.31 | 0.38 | 0.31 | 0.44 | 0.43 | 0.37 | 0.27 | 0.39 |
| Kurtosis | 0.66 | 0.63 | 0.51 | 0.50 | 0.74 | 0.84 | 0.59 | 0.43 |

# F. Hyperparameter Configuration and Sensitivity Analysis

## F.1. Edge Threshold Sensitivity

We evaluate the sensitivity of SMGD to the graph sparsification threshold used during brain graph construction. Specifically, we vary the proportion of retained strongest edges from 10% to 25% and report the results on both ABIDE-I and SRPBS-SSD. As shown in Table 7, SMGD remains stable within this range, with the best overall performance obtained at the 20% threshold used in our main experiments. This observation suggests that SMGD is not overly sensitive to the exact sparsification level, while a moderate threshold provides a favorable balance between preserving informative brain connectivity patterns and suppressing noisy edges.

*Table 7.* Sensitivity analysis of the graph sparsification threshold. All metrics are reported as mean $\pm$ standard deviation.

| Threshold | ABIDE-I | | | SRPBS-SSD | | |
|---|---|---|---|---|---|---|
| | ACC | F1 | MCC | ACC | F1 | MCC |
| 10% | 0.731±0.046 | 0.707±0.054 | 0.469±0.095 | 0.908±0.054 | 0.912±0.047 | 0.827±0.095 |
| 15% | 0.743±0.032 | **0.728±0.036** | 0.486±0.064 | 0.946±0.035 | 0.946±0.037 | 0.892±0.067 |
| 20% | **0.748±0.034** | 0.725±0.053 | **0.507±0.068** | **0.954±0.032** | **0.957±0.024** | **0.906±0.051** |
| 25% | 0.735±0.028 | 0.717±0.041 | 0.470±0.058 | 0.935±0.022 | 0.936±0.023 | 0.872±0.044 |

## F.2. Objective Hyperparameter Sensitivity

The total training objective in Eq. (16) combines the structural and diagnostic constraints of SMGD. We now detail each component and its internal weighting coefficients.

Following Eq. (9), the non-degeneracy loss ensures expressive, decorrelated representations within each subspace by jointly maximizing diagonal variances and minimizing off-diagonal covariances:

$$\mathcal{L}_{\text{nondeg}} = \sum_{\mathcal{A} \in \{S,C,I,N\}} \sum_{v \in \{v_1,v_2\}} \left[ \sum_{i \neq j} \text{Cov}(Z_{\mathcal{A}}^{(v)})_{ij}^2 - \epsilon_1 \sum_j \log(\text{Cov}(Z_{\mathcal{A}}^{(v)})_{jj}) \right], \tag{55}$$

where the internal coefficient $\epsilon_1$ acts as a scaling factor for numerical balancing.

Following Eqs. (12) and (13), the cross-modal regularization loss enforces alignment for shared subspaces and independence for specific subspaces:

$$\mathcal{L}_{\text{modal}} = \sum_{\mathcal{A}_1 \in \{S,I\}} \mathcal{L}_{\text{align}}(Z_{\mathcal{A}_1}) + \epsilon_2 \sum_{\mathcal{A}_2 \in \{C,N\}} \mathcal{L}_{\text{decoup}}(Z_{\mathcal{A}_2}), \tag{56}$$

where the internal coefficient $\epsilon_2$ balances the magnitude of the two terms.

Following the formulation in Eq. (14), the diagnosis loss combines task sufficiency for the diagnostic core and shortcut suppression for the incidental and nuisance components:

$$\mathcal{L}_{\text{diag}} = \mathcal{L}_{\text{core}} + \epsilon_3 \mathcal{L}_{\text{short}}, \tag{57}$$

where the internal coefficient $\epsilon_3$ balances maximizing $I(Z_{\text{core}}; Y)$ against minimizing $I(Z_{\text{short}}; Y)$.

The SMGD objective involves two levels of hyperparameters: (1) *primary hyperparameters* $\lambda_D$ and $\lambda_M$ that control the trade-off among the three main loss components, and (2) *internal balancing coefficients* $\epsilon_1, \epsilon_2, \epsilon_3$ that normalize constituent terms within each loss component. The internal coefficients are determined by the principle of loss magnitude balancing: since different loss terms may have vastly different scales, each coefficient is set to be approximately the reciprocal of the corresponding term's magnitude at initialization. Specifically, we compute the initial value of each raw loss on a held-out batch and set $\epsilon_i \propto 1/\mathcal{L}_i^{(0)}$. This ensures that all terms contribute roughly the same order of magnitude during early training, preventing any single term from dominating optimization.

Table 8 summarizes the hyperparameter configurations, including search ranges and optimal values determined via grid search on ABIDE-I. The corresponding sensitivity analysis is visualized in Figure 7, illustrating the impact of variations in these key hyperparameters on diagnostic performance.

*Table 8.* Hyperparameter configurations for sensitivity analysis. The optimal value is selected based on the highest mean accuracy.

| Symbol | Description | Search Range | Optimal | Category |
|---|---|---|---|---|
| $d_{\text{latent}}$ | Latent dimension per subspace | {8, 16, 32, 64, 128, 256} | 64 | Architecture |
| $\lambda_M$ | Modality weight ($\mathcal{L}_{\text{modal}}$) | {1e0, 5e0, 1e1, 2e1, 5e1, 1e2} | 2e1 | Primary |
| $\lambda_D$ | Diagnosis weight ($\mathcal{L}_{\text{diag}}$) | {1e1, 2e1, 3e1, 5e1, 1e2, 2e2} | 1e2 | Primary |
| $\epsilon_1$ | Variance balance in $\mathcal{L}_{\text{nondeg}}$ | {1e-3, 1e-2, 5e-2, 1e-1, 5e-1, 1e0, 2e0} | 1e-3 | Internal |
| $\epsilon_2$ | Independence balance in $\mathcal{L}_{\text{modal}}$ | {1e1, 2e1, 5e1, 1e2, 5e2, 1e3} | 1e2 | Internal |
| $\epsilon_3$ | Shortcut balance in $\mathcal{L}_{\text{diag}}$ | {5e-2, 1e-1, 5e-1, 1e0, 2e0, 5e0, 1e1} | 1e0 | Internal |

## G. Interpretability Analysis

Following the protocol in the main text, we further extend ROI attribution from modality-level explanations to SMGD's four disentangled components, so as to examine which anatomical regions are associated with each factorized pathway. We use Integrated Gradients (IG) with a zero baseline and 30 integration steps. IG is computed with respect to the node feature matrix of each modality graph, and ROI importance is defined as the sum of absolute IG values across node features, yielding one score per ROI.

For modality-level attribution, we attribute the ASD prediction logit, yielding two modality-specific ROI score vectors. For component-level attribution, we attribute the activation energy of each component. Specifically, for $A \in \{S, C, I, N\}$ and subject $m$, we concatenate the two modality-branch embeddings and define the component energy as $e_{A,m} = \left\| [z_{A,m}^{(v_1)}; z_{A,m}^{(v_2)}] \right\|_2$, which yields four component-specific ROI attribution maps. ROI scores are computed per subject and then averaged over the ABIDE-I cohort to obtain group-level importance. For visualization, we normalize scores within each map, retain the Top-10 ROIs for each component, and project them onto the BNA-246 atlas. Gaussian smoothing and voxel-wise thresholding are applied only for clearer glass-brain visualization, as shown in Figure 8. Because $Z_I$ and $Z_N$ are excluded from the diagnostic classifier, their attribution maps should be interpreted as shortcut- or nuisance-related activations rather than direct diagnostic evidence.

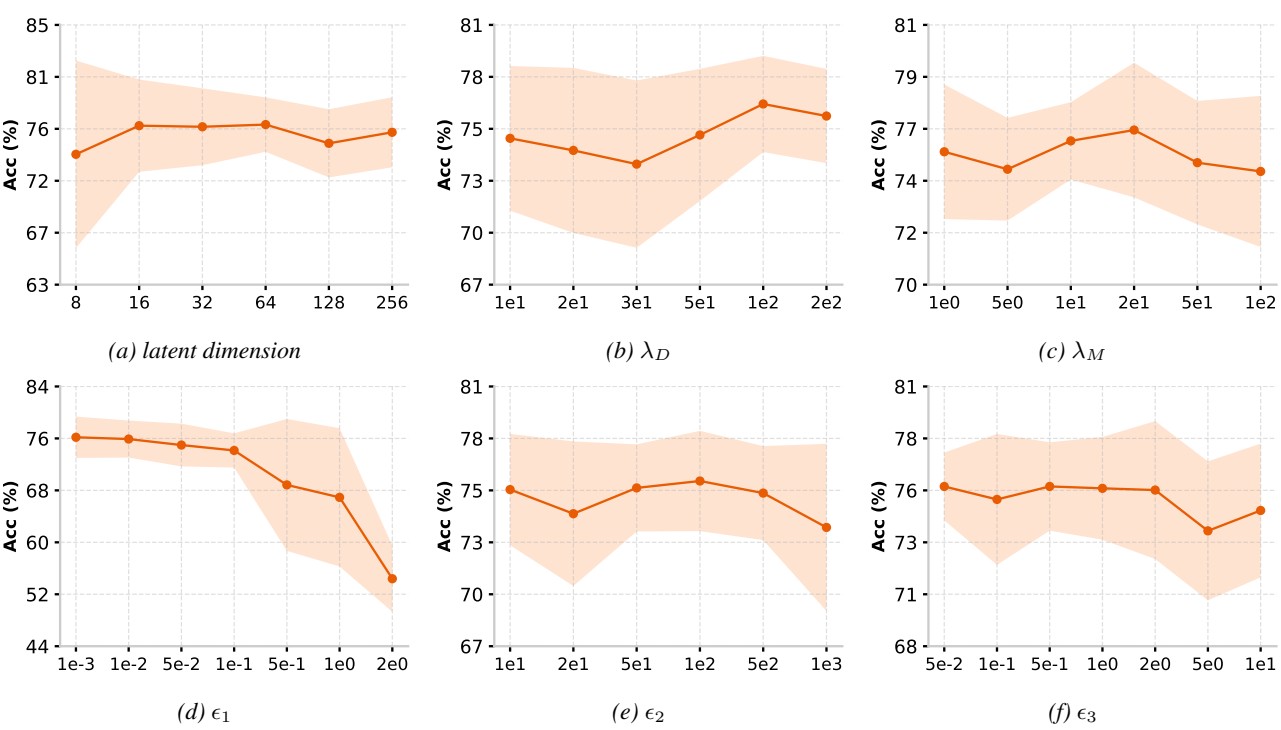

*Figure 7.* Sensitivity analysis of hyperparameters on the ABIDE-I dataset.

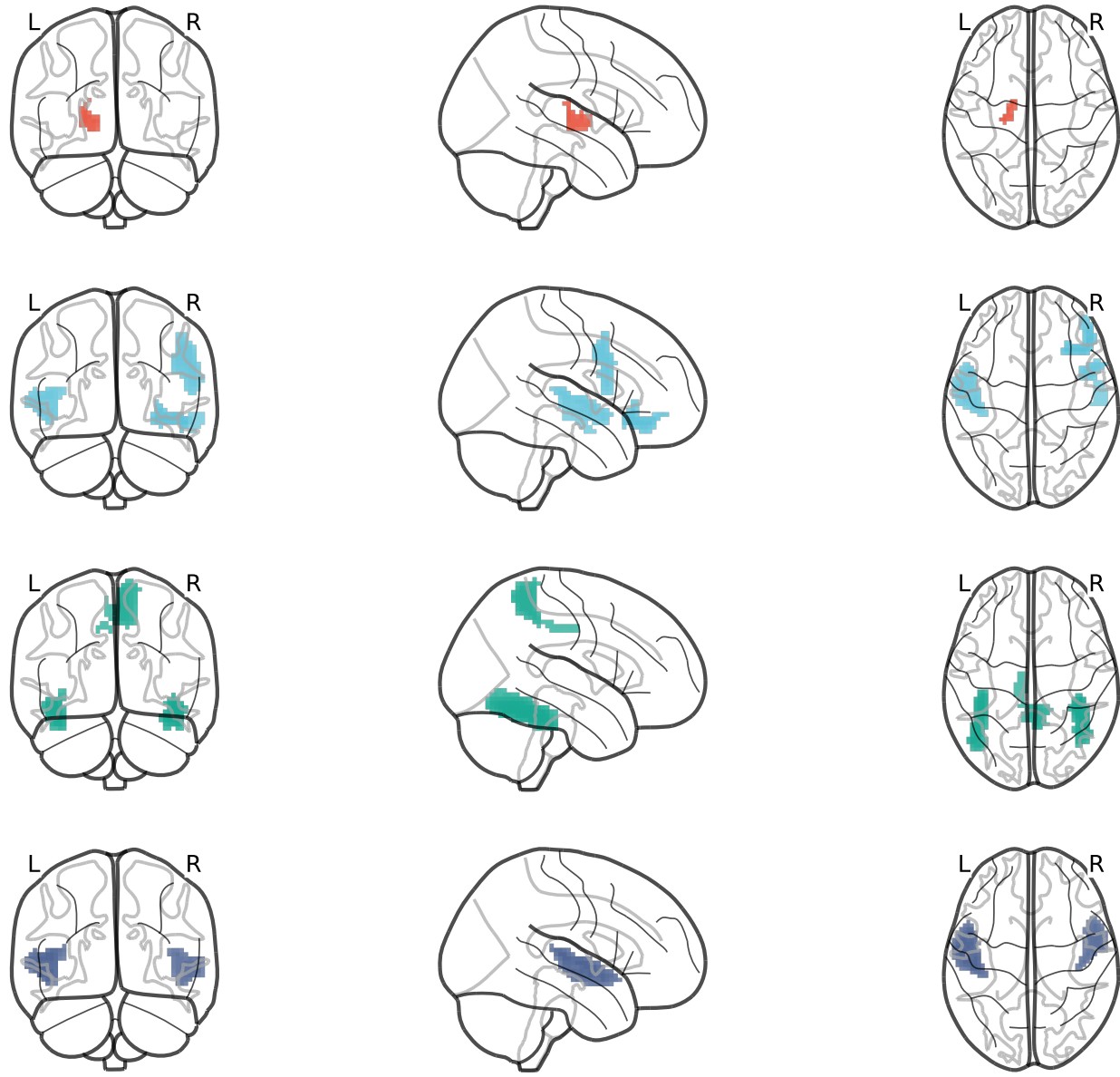

*Figure 8.* Subspace-wise glass-brain attribution maps for SMGD. Rows correspond to $Z_S$, $Z_C$, $Z_I$, and $Z_N$, shown (left to right) in coronal, sagittal, and axial views.

