# OpenReview forum: "Structured Multi-modal Graph Disentanglement for Psychiatric Diagnosis"
_ICML.cc/2026/Conference — ICML 2026 regular_

### Official Review · Reviewer_5eDE · 2026-03-05

**Soundness:** 3
**Presentation:** 3
**Significance:** 3
**Originality:** 3
**Overall Recommendation:** 4
**Confidence:** 3

**Summary:**

This paper proposes Structured Multi-modal Graph Disentanglement (SMGD), which is a novel framework for multi-modal neuroimaging diagnosis. The model factorizes multi-modal graph representations into four components: cross-modality shared diagnostic features, modality-specific features, incidental cross-modal features, and modality-specific non-robust correlations. Moreover, the model incorporates a mini-batch estimable regularizer that control relational geometry of modalities. Experiments are performed on ABIDE and SRPBS datasets.

**Compliance With Llm Reviewing Policy:**

Affirmed.

**Key Questions For Authors:**

Q1) Given the relatively small dataset size of medical data, there may be a risk of overfitting. Have any techniques been applied to avoid overfitting?

Q2) Since the geometric regularizer is applied to each batch, the batch size would affect the model performance. Ablation studies on the component $k$ would be helpful.

Q3) I would like to ask if authors will release the fully processed data used in the experiments.

Q4) How were the baseline methods fine-tuned? For example, have the authors tried to search for their optimal batch size and learning rates?

**Limitations:**

Limitations and future works are not discussed in the paper.

**Strengths And Weaknesses:**

Strength

$\bullet$ The motivation and rationale of the idea that divides multi-modal data into structured, four different representations are reasonable and novel. The loss functions of the proposed method technically align well with the idea.

$\bullet$ Extensive experiments were performed with various baseline methods across multiple datasets.

Weakness

$\bullet$ Both ABIDE and SRPBS datasets contain only two classes with limited sample sizes. In other words, simple binary classifications on control vs. patient groups are performed, which may limit the generalizability of the proposed method to more challenging real-world settings with multiple disease subtypes.

$\bullet$ The theoretical justification relies on a strong Gaussian assumption (DDH), which may not hold for deep neural representations in practice. Further discussion on the validity and applicability of the proposed method would be helpful.

---

> ### Author Rebuttal · Authors · 2026-03-29
>
> ### W1: Multi-class Generalizability
> Both our SRPBS dataset (encompassing multiple psychiatric disorders, see Appendix D) and the architectural implementation natively support multi-class classification tasks. We therefore validated our model by selecting the three most frequent disorders (255 MDD, 147 SSD, 125 ASD) alongside 791 healthy controls for a four-class classification task (covering 93.4% of the dataset), with results presented below:
> |Metric|GCN|**SMGD**|
> |--------|-----|------|
> |ACC|.768±.040|**.795±.043**|
> |Balanced ACC|.576±.068|**.614±.092**|
> |Macro F1|.695±.052|**.719±.079**|
>
> ### W2: Theoretical Applicability (DDH Boundaries)
> - **DDH Verification.** In response to Reviewer Mzom (Q1), we mentioned that both the original DDH paper and our work (Appendix E) have verified that hidden representations can be well approximated by Gaussian distributions across several classical models.
>
> - **Applicability of the method.** The dependence of SMGD's objective functions on DDH is as follows:
>
> |Loss Term|Explanation|
> |---|---|
> |$\mathcal{L}_{nondeg}$ (Eq.9)|Proxy for entropy|
> |$\mathcal{L}_{modal}$ (Eq.12, Eq.13)|$I(Z_{decoup}^{(V_1)}; Z_{decoup}^{(V_2)}) \to \text{Cov}(Z_{decoup}^{(V_1)}; Z_{decoup}^{(V_2)})$|
> |$\mathcal{L}_{diag}$(Eq.14)|$I(Z_{short}; Y) \to \text{Cov}(Z_{short}; Y)$|
>
> We therefore first clarify that the theoretical analysis of SMGD's robustness to the modality gap (Appendix B.4) does not depend on this assumption. We then derive the applicable boundary conditions for the three terms: Since DDH approximates MI as covariance, the approximation error $\Delta E = I(A;B) - O(\lVert\text{Cov}(A,B)\rVert_F^2)$ will become unbounded in three extreme distribution scenarios:
>
>  **1.Heavy-Tailed Distributions.** Regarding $\mathcal{L}\_{nondeg}$, using variance as a surrogate for entropy fails under these conditions, as the model can inflate variance by pushing a large number of discrete outliers to extreme values (i.e., $\Delta E \to \infty$), while the true MI remains very low (Figure 3 illustrates this scenario).
>
> **2.Extreme Non-Linear Correlations.** When the dataset exhibits severe systematic biases that drive the model to learn extreme high-order non-linear correlations (e.g., $Z_C^{v_1}={Z_C^{v_2}}^2$), the resulting divergence in approximation error renders the decoupling constraint $\mathcal{L}\_{\text{decoup}}$ ineffective.
>
> **3.Non-Unimodal Distributions.** For $\mathcal{L}\_{short}$, zero-covariance alignment can still suffer from information leakage due to intra-class heterogeneity under multimodal distributions.
>
> Overall, when the data exhibits strong non-linear dependence or extreme outliers, the DDH-based surrogate objectives will collapse. We will add these to the Limitations section.
>
> ### Q1: Strategies to Prevent Overfitting
> - In the field of brain network analysis, typical benchmark datasets usually range from 100 to 1,000 samples. By comparison, ABIDE-I and SRPBS are representative large-scale multi-site datasets within the field.
> - We have incorporated mechanisms to mitigate overfitting throughout the entire pipeline, from initial data preprocessing to the disentangled representation in our framework design (see Appendix D and our response to Reviewer TCPP Q2).
> - We employ rigorous stratified cross-validation for model evaluation and result reporting. Crucially, Table 2 demonstrates robust cross-dataset generalization against distribution shifts, confirming the model captures site-invariant pathological features rather than overfitting to specific datasets.
>
> ### Q2: Batch Size Sensitivity
> While SMGD depends on batch size, evaluations on two datasets confirm stability within a moderate range:
> |Batch size|ABIDE (ACC)|ABIDE (F1)|ABIDE (MCC)|SRPBS (ACC)|SRPBS (F1)|SRPBS (MCC)|
> |------------|-----------|----------|-----------|-----------|----------|-----------|
> |8|.715±.047|.678±.064|.427±.083|.912±.063|.912±.064|.825±.126|
> |16|.735±.032|.708±.055|.475±.064|.932±.036|.936±.030|.872±.064|
> |32(used)|.748±.034|.725±.053|.507±.068|.954±.032|.957±.024|.906±.051|
> |64|.733±.041|.713±.064|.474±.085|.893±.049|.889±.063|.789±.093|
>
> ### Q3: Data Availability
> Adhering to the usage agreements and ethical guidelines of the original data consortium, we have provided access to all SRPBS data via the original anonymous link (folder "SRPBS_"); ABIDE-I will be released upon publication.
>
> ### Q4: Baseline Comparisons
> For fair comparison, we used a unified hyperparameter grid search (e.g., lr, GNN layers) for all baseline models; for model-specific hyperparameters (e.g., the trade-off parameter in GIB, channel numbers in DGCL), the grid search intervals were concentrated in the optimal value regions reported in the original papers; for certain works adapted to the ABIDE/SRPBS datasets (e.g., CI-GNN, BioBGT), we adopted the published configuration parameters.
>
> ### Discussion on Limitations and Future Work
> We appreciate the reviewer’s suggestion, we will add these sections in the revised manuscript.

---

> > ### Author Rebuttal · Reviewer_5eDE · 2026-04-03
> >
> > Most of my concerns were addressed. I will maintain the original score, which is already positive.

---

> > > ### Author Response · Authors · 2026-04-03
> > >
> > > Thank you for your constructive feedback and for acknowledging that our rebuttal resolved your concerns. We greatly appreciate the positive score, and we hope our detailed responses will be weighed favorably in your final assessment of our work.

---

### Official Review · Reviewer_Mzom · 2026-03-05

**Soundness:** 3
**Presentation:** 3
**Significance:** 3
**Originality:** 3
**Overall Recommendation:** 5
**Confidence:** 3

**Summary:**

This paper proposes a Structured Multimodal Graph Decomposition (SMGD) framework for multimodal neuroimaging diagnosis. The authors attempt to forcefully decompose complex brain network features into four distinct subspaces with clear medical significance (shared disease, complementary disease, incidental noise, and specific noise). The goal is to enable predictions based solely on pure disease features, filtering out environmental interference such as hospital equipment.  Extensive experiments demonstrate that its algorithm outperforms current methods on the ABIDE-I and SRPBS-SSD datasets.

**Compliance With Llm Reviewing Policy:**

Affirmed.

**Final Justification:**

The author resolved most of my issues, so I've upgraded my score to 5 .

**Key Questions For Authors:**

See Weakness

**Limitations:**

yes

**Strengths And Weaknesses:**

Strengths:

1. This paper is clearly written and well structured.

2. Extensive experiments demonstrate that its algorithm outperforms current methods on the ABIDE-I and SRPBS-SSD datasets.

3. The theoretical proofs are abundant and well-grounded.

Weakness：
1. The Data Distribution Hypothesis requires simple piecewise affine networks (e.g., ReLU), but the model architecture adopted in subsequent experiments appears to have deviated from this assumption.

2. In terms of parameter sensitivity, the model exhibits significant sensitivity to parameters. Does the resource consumption resulting from parameter tuning limit its practical application?

3. The paper decomposes features into four subspaces. How can we prove the success of decoupling? Should we remain skeptical about the success of decoupling? Is the improvement merely due to increased parameters and tuning?

4. Although the article proposes decoupling into four parts, it essentially distinguishes between task-irrelevant and task-relevant components. So what is the significance of further subdividing the task-irrelevant parts?

---

> ### Author Rebuttal · Authors · 2026-03-29
>
> ### W1: Applicability of DDH
>
> We answer this from both theoretical and empirical perspectives:
>
> - **Theoretical View**: In the original paper proposing DDH (Shwartz-Ziv et al., 2023), although the initial derivation relies on (leaky-)ReLU, the authors subsequently extend their conclusion to non-piecewise affine smooth activation functions in Section 2.1:
> >*"For smooth nonlinearities, our results hold using a first-order Taylor approximation argument."*
>
> This guarantees that SMGD can preserve representation Gaussianity despite utilizing non-piecewise affine operators.
>
> - **Post-hoc Stress Test**: In Figure 1 of DDH, the authors conducted experiments using backbones such as SimCLR and SwAV. Neither is strictly piecewise affine, yet DDH still approximately holds. In our paper, we empirically validate DDH through a normality analysis (Appendix E). The standard Skewness and Kurtosis of SMGD's representations confirm Gaussianity that would not manifest if the architecture severely violated DDH.
>
> ### W2: Parameter Sensitivity and Tuning Cost
>
> - The model is insensitive to most hyperparameters (HPs) (e.g., $\lambda_M$, $\lambda_D$, $\varepsilon_2$ and $\varepsilon_3$) as shown in Figure 9, yielding performance remains stable across 1 to 2 orders of magnitude. The sensitivity of $\varepsilon_1$ stems from an inherent magnitude gap of approximately $10^3$ between the two terms in Eq.9; setting it too large causes model degradation. We address this via a magnitude-balancing initialization strategy (Appendix F), requiring no HP tuning.
>
> - After applying the aforementioned initialization strategy, only $\lambda_M$ and $\lambda_D$ actually require tuning, each with six candidate values. This scale is significantly smaller than the search space of baseline methods like BrainOOD and CI-GNN.
>
> - As described in Section 5.1, all HPs determined on ABIDE-I were directly applied to SRPBS-ASD without any re-tuning. SMGD still achieved performance improvements, indicating that the selected HPs do not rely on a single dataset. This demonstrates that SMGD's HPs are robust in practical deployment.
>
> ### W3: Decoupling Verification and Model Complexity
>
> **Validation of Disentanglement**
>
> - Figure 8 (Appendix E) reports both covariance-based linear and HSIC-based nonlinear dependencies. Results show that the HSIC values between cross-modal/cross-type subspace pairs (e.g., $Z_{C}^{(v1)}$vs.$Z_{S}^{(v2)}$) are significantly lower than those within the same subspace category, exhibiting a clear block-wise disentanglement structure.
>
> - As shown in Figure 10, the four subspaces exhibit distinct ROI activation patterns, suggesting successful decoupling rather than overlapping representations.
>
> - In our response to Reviewer TCPP (Q2), we supplemented linear probe results for four representations, empirically demonstrating that the representations successfully decouple diagnostic from nuisance information represented by age.
>
> **Benchmarking Parameter Scale**
>
> We summarize the parameter counts and the number of architecture-specific HPs for representative models across categories in Table 1:
> ||CIGNN|GAT|IBGNN|BioBGT|BrainOOD|**SMGD**|
> |:---|:---|:---|:---|:---|:---|:---|
> |**Parameters**|289K|519K|762K|943K|1.13M|1.36M|
> |**HPs**|3|1|1|1|4|2 (+3 Tuning-free)|
>
> While our model has a slightly higher parameter count, it is not large enough to yield a significant scale effect. Furthermore, Table 4 shows that ablating SMGD ("w/o SMGD", retaining a comparable parameter count via basic encoder and fusion) reduces performance from 0.748 to 0.703. This confirms our gains mainly stem from structural regularization rather than parameter scale.
>
> ### W4: Motivation for Finer Subspaces
>
> This relates to a seeming paradox in Table 4: theoretically, since $Z_I$/$Z_N$ bypass task optimization, ablating their constraints should not affect performance, yet it leads to severe degradation.
>
> We attribute this to the following mechanism: SMGD jointly constrains representations along two dimensions— **task** and **modality**. Specifically, our model architecture first separates diagnostic core from shortcut nuisances (the task axis), and then further decomposes each into shared and specific components (the modality axis), as detailed in Appendix C. This hierarchical structure implies that the modality-axis constraints act symmetrically on both subspace pairs $Z_S$&$Z_C$ and $Z_I$&$Z_N$. Consequently, if $Z_I$&$Z_N$ are left undifferentiated, the model's structural control over the latent space would significantly diminish, making it unable to decouple the corresponding structural noise from $Z_S$&$Z_C$, which ultimately degrades downstream task performance.
>
> The degradation observed in the "w/o $Z_I$/$Z_N$" setting thus serves as a direct reflection of this: by structurally distinguishing $Z_I$ from $Z_N$, the model anchors the structural identity of these subspaces, which in turn compels confounding factors to be correctly disentangled from $Z_S$ and $Z_C$.

---

> > ### Author Rebuttal · Reviewer_Mzom · 2026-04-02
> >
> > Thank you for the authors' reply. I will keep my positive score.

---

> > > ### Author Response · Authors · 2026-04-03
> > >
> > > Thank you once more for your careful follow-up and for acknowledging the full resolution of your concerns. As the score was submitted before we had the opportunity to address these points, we would be honored if these clarifications could be taken into account during your final assessment of our work. We respect your decision to keep the rating as is, but we would be very appreciative of any potential reconsideration. Thank you once more for your time and for the constructive feedback.

---

### Official Review · Reviewer_hN8j · 2026-03-12

**Soundness:** 3
**Presentation:** 3
**Significance:** 2
**Originality:** 3
**Overall Recommendation:** 4
**Confidence:** 4

**Summary:**

This paper tackles multi-modal neuroimaging diagnosis by disentangling modality-shared, modality-specific, diagnostic-relevant, and diagnostic-agnostic information. To enable disentanglement while estimating mutual information in a closed form, the framework is built on the Data Distribution Hypothesis (DDH). Based on this formulation, the authors propose Structured Multi-modal Graph Disentanglement (SMGD). Diagnostic sufficiency and neutrality are encouraged through mutual information constraints with respect to the label, while modality commonality and specificity are enforced via distributional alignment and by restricting off-diagonal covariance terms.

**Compliance With Llm Reviewing Policy:**

Affirmed.

**Final Justification:**

I have read the rebuttal and updating my score accordingly.

**Key Questions For Authors:**

- Please see the weakness.
- What would be the result if $Z_S^{(v1)}$ and $Z_S^{(v2)}$ are exchanged across modalities? Since the framework does not impose pairwise alignment at the sample level, it is unclear whether distributional alignment together with task sufficiency would lead to semantically aligned representations across modalities, even if they are not identical.

**Limitations:**

Regarding W1 and W4, a discussion on scenarios involving more than two modalities would be valuable. Otherwise, the paper should clearly specify that the proposed framework is primarily designed for dual-modality settings. We look forward to seeing clarification on this point in the authors’ response.

**Strengths And Weaknesses:**

Strengths:
- The paper provides strong empirical validation to support its hypotheses. The alignment between interpretability analysis and improved predictive performance further strengthens the credibility of the proposed framework.

- The paper is generally well-written and easy to follow. To support the proposed framework and its underlying hypotheses, the authors design appropriate experiments that provide empirical validation.

- The paper addresses an important problem in psychiatric diagnosis. Given that multi-site and small-sample datasets are common in medical domains, the proposed approach has the potential to benefit such settings.

- The framework is designed to reflect the proposed paradigm through two key aspects, task relevance and modality commonality. This design is plausible in dual-modality scenarios and is well incorporated into the model architecture.

Weakness:
- W1: According to Partial Information Decomposition (PID), it is natural to consider the existence of synergistic information, which can only be accessed when multiple modalities are jointly available. It is unclear whether the current framework can accommodate such synergistic information when more than two modalities are involved.

- W2: The tolerances for structural consistency and independence (ε) are not clearly specified. No concrete values or sensitivity analyses are provided. It would be helpful to understand whether introducing such tolerances is more beneficial than directly minimizing or maximizing the corresponding objectives.

- W3: In the Implementation Details section, GPS is introduced without citation or explanation. The explanation appears later in the Pipeline Architecture section, which may make it difficult for readers to follow. Similarly, “Integrated Gradients” appears without an accompanying description or citation.

- W4: The framework may face limitations when extended to settings with more than two modalities. Since the method involves covariance-based computations and Wasserstein-based alignment, the computational complexity is not discussed. As the current experiments focus on dual-modality diagnosis and the framework does not include domain-specific psychiatric components, additional validation on datasets with more modalities (e.g., neurodegenerative datasets such as ADNI) would strengthen the generality of the approach.

---

> ### Author Rebuttal · Authors · 2026-03-31
>
> ### W1&W4: Multi-modal Extension
>
> PID provides a closed-form atomic decomposition for two modalities; however, in the tri-modal scenario, the number of atoms surges to 18 (Lyu et al., 2026), with complexity exploding exponentially as dimensionality increases. There is currently no universally recognized solution for this, and it remains an open problem in information theory. Consequently, most PID-inspired models target dual-modal scenarios, SMGD also faces this limitation. We will clearly articulate this theoretical bottleneck as a limitation and outline it as a key avenue for subsequent studies.
>
> However, we can draw inspiration from the pairwise interaction analysis used in [1] when handling tri-modal PID (by running dual-modal analyses separately on each modality pair, then averaging the estimates of the three pairs as the overall estimate) to implement an engineering approximation. We perform standard SMGD on all modality pairs separately to obtain three sets of diagnostic cores $Z_{core}^{12}$, $Z_{core}^{13}$, and $Z_{core}^{23}$ (superscript $i$ denotes modality $v_i$ for brevity), and then feed all cores into the classifier. The total modality loss becomes the sum of the three pairs, while the rest remains unchanged.
>
> We conducted the tri-modal binary classification experiment on the ADNI dataset (285 EMCI, 172 LMCI and demographic-matched NC), adding the fractional Amplitude of Low-Frequency Fluctuations (fALFF) modality. All other modalities and settings follow the SRPBS configuration. The results are:
> ||ACC|F1|MCC|
> |---|---|---|---|
> |NC vs. EMCI|.765±.045|.782±.039|.536±.088|
> |NC vs. LMCI|.779±.065|.791±.040|.581±.107|
>
> While this does not capture 3rd-order synergy as W1 accurately pointed out (a common PID limitation), it proves SMGD's empirical scalability.
>
> ### W2: Justification for Tolerances
> - In constrained information-theoretic representation learning, using tolerance constraints rather than strict optimization objectives is widely adopted. For example, FACTORCL (Liang et al., NIPS, 2023) defines multi-view redundancy as $I(X_1;Y|X_2)≤\epsilon$. At the implementation level, these tolerances are subsequently absorbed as hyperparameters through Lagrangian relaxation. The sensitivity analysis of $\lambda_M$, $\lambda_D$, and $\epsilon_i$ in Appendix F is actually an empirical evaluation of them.
> - In the context of this paper, the necessity of the two tolerances is as follows: For structural consistency (Eq. 5), the tolerance is the guarantee that allows for the existence of centroid shifts between modalities; for shortcut neutrality (Eq. 4), if $I(Z_{short};Y)\to0$ is strictly required, the gradient will drive $Z_{short}$ to degenerate into a constant representation.
>
> ### W3: Clarity Terminology
>
> We thank the reviewer for pointing this out. We will properly define/cite them at their initial mention in the revised manuscript.
>
> ### Q1: Assessment of Alignment
> - **Statistical Evidence.** CKA analysis (a standard representation similarity metric) in Figure 8a shows significantly higher similarity for shared $Z_S$/$Z_I$ than for modality-specific $Z_C$/$Z_N$, providing statistical evidence for cross-modal alignment.
> - Procrustes Analysis (Added). We added a zero-shot feature-swapping inference experiment on the frozen model:
> |Configuration|ACC|F1|MCC|
> |:---|:---|:---|:---|
> |$[Z_C^{1}, Z_C^{2}, Z_S^{1}, Z_S^{2}]$ (Baseline)|.748±.034|.725±.053|.507±.068|
> |Swap $Z_S^{1}$ and $Z_S^{2}$|.499±.029|.402±.207|.015±.089|
> |Swap $Z_C^{1}$ and $Z_C^{2}$|.691±.043|.703±.038|.448±.063|
>
> Swapping $Z_S^{1}$ and $Z_S^{2}$ initially drops performance. Since GW alignment is invariant to rigid transformations, we hypothesize these representations remain topologically isomorphic, the degradation may reflect the coordinate-sensitive classifier's failure on unaligned bases. To verify this, we introduced Orthogonal Procrustes Analysis [2], which extracts the training-set features $Z_S^{1}$ and $Z_S^{2}$ and performs SVD to find the optimal transformation matrix/vector $R$ and $t$ that minimizes the Frobenius distance between the two representations (i.e., their mapping relationship can be expressed as $\hat{Z}_S^{1}=Z_S^{2} \cdot R+t$, and vice versa). We then replace the original representations on the test set with the transformed proxies (denoted $\hat{Z}$) then achieved near-baseline restoration ($Z_S^{2}$ shows partial recovery):
> |Configuration|ACC|F1|MCC|
> |:---|:---|:---|:---|
> |Replace $Z_S^{1}$ by $\hat{Z}_S^{1}$|.738±.031|.720±.039|.479±.061|
> |Replace $Z_S^{2}$ by $\hat{Z}_S^{2}$|.598±.037|.573±.178|.215±.069|
>
> This confirms SMGD embeds consistent semantics in topologically isomorphic but coordinate-shifted systems, aligning with LLM translation findings [2].Code for W1&Q1 has been uploaded to the original anonymous link.
>
> Reference
>
> [1] Efficient Quantification of Multimodal Interaction at Sample Level. ICML,2025
>
> [2] Representational Isomorphism and Alignment of Multilingual Large Language Models. EMNLP,2024

---

> > ### Author Rebuttal · Reviewer_hN8j · 2026-04-03
> >
> > Thanks for the comments and I have no further questions.

---

> > > ### Author Response · Authors · 2026-04-03
> > >
> > > We are delighted to know that our rebuttal and the additional clarifications have successfully addressed your concerns. We sincerely thank you for your constructive feedback and the effort you spent on our work.

---

### Official Review · Reviewer_TCPP · 2026-03-13

**Soundness:** 4
**Presentation:** 4
**Significance:** 3
**Originality:** 3
**Overall Recommendation:** 4
**Confidence:** 4

**Summary:**

The authors correctly argue that many multimodal diagnostics fail to disentangle generalizable cross-modal signals from modality-specific effects. To address this issue, they propose a new framework, Structured Multimodal Graph Disentanglement (SMGC), which decomposes patterns into four components: shared diagnostic signals, complementary diagnostic signals, incidental consensus, and modality-specific nuisances. They implement their method for psychiatric disorder diagnosis.

**Compliance With Llm Reviewing Policy:**

Affirmed.

**Final Justification:**

The rebuttal partially addressed my concerns, so I changed my score from 3 to 4.

**Key Questions For Authors:**

- How did you decide on the top 20% strongest edges for your adjacency matrix, and how sensitive are the results to this choice?
- How does your model account for subject-to-subject variability, which is common in fMRI data?

**Limitations:**

There are no deep theoretical contributions; the theoretical discussion primarily relies on existing results.

**Strengths And Weaknesses:**

Strengths
- The idea of imposing structure on the latent space to identify different patterns is interesting and, in this context, novel.
- The authors provide a step-by-step process for defining the four subspaces and clearly motivate their choices.
- The authors provide a very thorough evaluation of their method.
- The experiments are extensive and well-developed, not an afterthought.

Weaknesses
- The theoretical parts of the paper mainly rely on standard, relatively straightforward results.
- While the method is conceptually very interesting, from a methodological perspective it may appear incremental relative to Partial Information Decomposition.

---

> ### Author Rebuttal · Authors · 2026-03-29
>
> ### **Q1: Edge Threshold Sensitivity**
>
> Thanks for asking about thresholds. Our decision was primarily driven by existing literature: (1) **BrainGB** [1] demonstrated that Proportional Thresholding better aligns the topological structures across different subjects, significantly enhancing GNN generalization. (2) Prior works such as **GAT-LI** [2] and **NeuroGraph** [3] conducted large-scale benchmark experiments on edge sparsity in GNN models, indicating that edge sparsity impacts downstream tasks in a mild inverted-U trend, with a stable performance plateau within the 10%–30% range. Therefore, we selected 20% as a robust empirical prior. To address this concern, we conducted comprehensive ablation experiments:
>
> |Threshold|ABIDE(ACC)|ABIDE(F1)|ABIDE(MCC)|SRPBS(ACC)|SRPBS(F1)|SRPBS(MCC)|
> |----------|-----------|----------|-----------|----------|---------|----------|
> |10%|.731±.046|.707±.054|.469±.095|.908±.054|.912±.047|.827±.095|
> |15%|.743±.032|.728±.036|.486±.064|.946±.035|.946±.037|.892±.067|
> |20% (used)|.748±.034|.725±.053|.507±.068|.954±.032|.957±.024|.906±.051|
> |25%|.735±.028|.717±.041|.470±.058|.935±.022|.936±.023|.872±.044|
>
> Results are consistent with literature, and we will include this detailed analysis in Appendix F.
>
> ### **Q2: Inter-subject Variability**
>
> We appreciate the reviewer highlighting this common fMRI challenge. Actually, One primary motivation of SMGD is to disentangle subject-to-subject variability, which is treated as spurious correlations in our manuscript and to mitigate it through four complementary levels:
>
> - Preprocessing: Standard DPABI pipelines and 20% edge thresholding act as a low-pass filter to remove weak, unstable individual connectivity fluctuations.
> - Model Mechanism: The quadruple factorization actively isolates variability into statistically independent nuisance subspaces ($Z_I, Z_N$), keeping the diagnostic core unaffected.
> - Generalization (Table 2): Extreme multi-site heterogeneity (ABIDE-I vs. SRPBS-ASD) would cause models overfitted to individual variations to inevitably collapse. Our robust zero-shot generalization empirically proves the successful decoupling of these nuisances.
> - Linear Probe (Added): We added a linear probe experiment to predict age on ABIDE-I, training on representations extracted from the penultimate layer of the frozen model. The results prove individual confounds are effectively trapped in nuisance components ($R^2=0.255$) rather than the diagnostic core ($R^2=0.087$):
>
>   |Index|$Z_C$|$Z_S$|$Z_I$|$Z_N$|$Z_C+Z_S$|$Z_I+Z_N$|
>   |-----|---------|---------|---------|---------|--------------|-------------|
>   |$R^2$|.025±.050|.098±.140|.202±.141|.411±.159|**.087 ±.120**|**.255±.185**|
>
> ### **W1: Theoretical Contribution**
>
> Our novelty lies in mathematically formalizing critical domain failure modes:
>
> - Problem Formalization in a Non-trivial Scenario. Contrastive learning and multimodal large models in generic domains also suffer from the Modality Gap and Spurious Correlations; however, these issues are largely masked in image-text scenarios (e.g., aligning a dog's photo with the word 'Dog' easily distills clear semantics). In medical imaging, however, the strongest signals are often site-specific noise rather than true pathology. Therefore, we formalize this specific issue to mandate finer-grained control over information structures and provide a foundation for multimodal representation disentanglement.
>
> - Mathematical proof of failure modes. Rather than merely observing the empirical failures of standard models and heuristically patching them, we leverage representation disentanglement and information theory to mathematically prove the inherent failure modes of these paradigms (see Appendix B), and ultimately unify the standard fusion/alignment pipeline into a rigorous information-theoretic framework.
>
> ### **W2: Comparison with PID**
>
> We appreciate the insightful comment. We clarify that SMGD is not merely an incremental PID by examining their core motivations and application scopes:
>
> - PID is general and task-centric, which means it cannot model and decouple the aforementioned spurious correlations, whereas SMGD is representation-centric and specifically targets this issue.
> - PID is fundamentally a descriptive tool in information theory typically used for post-hoc analysis to quantify information proportions. In contrast, SMGD is a complete constrained optimization framework designed to actively optimize representations for the specific characteristics of medical imaging data.
>
> For further conceptual and component-wise comparisons, please refer to Appendix A.
>
> ### Reference
>
> [1] BrainGB: A Benchmark for Brain Network Analysis with Graph Neural Networks. TMI,2022
>
> [2] GAT-LI: A Graph Attention Network Based Learning and Interpreting Method for Functional Brain Network Classification. BMCB,2021
>
> [3] NeuroGraph: Benchmarks for Graph Machine Learning in Brain Connectomics. NIPS,2023

---

> > ### Author Rebuttal · Reviewer_TCPP · 2026-04-03
> >
> > Thanks for the comments, and I have no further questions. I adjusted your score accordingly.

---

> > > ### Author Response · Authors · 2026-04-04
> > >
> > > ​Thank you for the supportive comments and for confirming that our response alleviated your concerns. We will ensure that your insightful recommendations are fully incorporated into the final revision to further strengthen our work.

---

### Decision · Program_Chairs · 2026-04-30

**Decision:**

Accept (regular)

**Comment:**

This paper studies multimodal neuroimaging diagnosis and proposes SMGD, a structured disentanglement framework with strong empirical performance on multi-site psychiatric datasets. The reviewers generally agreed that the paper addresses an important problem and that the proposed factorization is both novel and well motivated, with thorough experiments and promising cross-dataset generalization. The rebuttal also resolved the main concerns raised in the initial reviews, including sensitivity to the graph threshold and batch size, clarification of the role and limits of the DDH assumption, empirical support for the intended disentanglement, and additional discussion of multi-class and approximate multi-modal extensions. The remaining issues are relatively minor and mostly concern scope rather than correctness, such as the fact that the current formulation is primarily grounded in dual-modality settings and that the theoretical contribution is less strong than the empirical one. Overall, the paper received consistently positive reviewer feedback after rebuttal, and I recommend acceptance (poster)